# Integrated annotation prioritizes metabolites with bioactivity in inflammatory bowel disease

Amrisha Bhosle [1,2,3], Sena Bae [4], Yancong Zhang [1,2,3], Eunyoung Chun [4], Julian Avila-Pacheco [5], Ludwig Geistlinger [6,7], Gleb Pishchany [1], Jonathan N Glickman [8,9], Monia Michaud [4], Levi Waldron [6], Clary B Clish [5], Ramnik J Xavier [1,10,11], Hera Vlamakis [1], Eric A Franzosa [1,2,3,13], Wendy S Garrett [1,3,4,12,13] & Curtis Huttenhower [1,2,3,4,13] ✉

## Abstract

**Microbial biochemistry is central to the pathophysiology of inflammatory bowel diseases (IBD). Improved knowledge of microbial metabolites and their immunomodulatory roles is thus necessary for diagnosis and management. Here, we systematically analyzed the chemical, ecological, and epidemiological properties of ~82k metabolic features in 546 Integrative Human Microbiome Project (iHMP/HMP2) metabolomes, using a newly developed methodology for bioactive compound prioritization from microbial communities. This suggested >1000 metabolic features as potentially bioactive in IBD and associated ~43% of prevalent, unannotated features with at least one well-characterized metabolite, thereby providing initial information for further characterization of a significant portion of the fecal metabolome. Prioritized features included known IBD-linked chemical families such as bile acids and short-chain fatty acids, and less-explored bilirubin, polyamine, and vitamin derivatives, and other microbial products. One of these, nicotinamide riboside, reduced colitis scores in DSS-treated mice. The method, MACARRoN, is generalizable with the potential to improve microbial community characterization and provide therapeutic candidates.**

**Keywords** Bioactive Compound Prioritization; Computational Method; Metabolomics; Microbiome; Inflammatory Bowel Disease
**Subject Categories** Metabolism; Methods & Resources; Microbiology, Virology & Host Pathogen Interaction

## Introduction

Microbial communities utilize, synthesize, and exchange metabolites for growth, sustenance, and response to environmental fluctuations. Microbial metabolism also contributes to environmental modification, a phenomenon central to bioremediation and nutrient-cycling (Jansson and Hofmockel, 2020; Kour et al, 2021), as well as to the health and disease-associated roles of the microbiome in a host (McCarville et al, 2020). The resulting chemical interactions are thought to be especially rich due to co-evolution (Baquero and Nombela, 2012; Rosenberg and Zilber-Rosenberg, 2018), and the human microbiome in particular produces metabolites that participate in processes including regulatory signaling, host metabolism, and modulation of the immune system (Krautkramer et al, 2021; McCarville et al, 2020). Stool metabolomics has thus become an elegant tool to study the molecular dialog underlying host–microbe and microbe–microbe interactions (Zierer et al, 2018). Comparative metabolomics of germ-free (GF) and conventional mice show that a remarkable ~10% of plasma metabolites are influenced by gut microbes (Wikoff et al, 2009), and it has been estimated that ~70% of the variation in human gut metabolomic profiles can be explained by differences in the gut microbiome (Zierer et al, 2018). However, of the tens of thousands of unique features detected in stool by high-resolution, untargeted mass spectrometry, only around a thousand are confidently identified (Lloyd-Price et al, 2019). This leaves the vast majority of stool metabolites uncharacterized (Wishart et al, 2022), with even fewer linked with specific microbial interactors or host phenotypic responses.

The subset of microbially derived gut metabolites that are characterized include several classes that are critically important in immunoregulation, energy harvest, and signaling: fatty acids, bile acids, lipids, amino acid derivatives, terpenoids, and oligosaccharides, among others (Donia and Fischbach, 2015; Han et al, 2021; Yen et al, 2015). These chemical classes contain at least some

[1] Infectious Disease and Microbiome Program, Broad Institute of MIT and Harvard, Cambridge, MA, USA. [2] Department of Biostatistics, Harvard T. H. Chan School of Public Health, Boston, MA, USA. [3] Harvard Chan Microbiome in Public Health Center, Harvard T. H. Chan School of Public Health, Boston, MA, USA. [4] Department of Immunology and Infectious Diseases, Harvard T. H. Chan School of Public Health, Boston, MA, USA. [5] Metabolomics Platform, Broad Institute of MIT and Harvard, Cambridge, MA, USA. [6] Department of Epidemiology and Biostatistics, Graduate School of Public Health and Health Policy, City University of New York, New York, NY, USA. [7] Center for Computational Biomedicine, Harvard Medical School, Boston, MA, USA. [8] Beth Israel Deaconess Medical Center, Boston, MA, USA. [9] Department of Pathology, Harvard Medical School, Boston, MA, USA. [10] Gastrointestinal Unit and Center for the Study of Inflammatory Bowel Disease, Massachusetts General Hospital and Harvard Medical School, Boston, MA, USA. [11] Center for Microbiome Informatics and Therapeutics, Massachusetts Institute of Technology, Cambridge, MA, USA. [12] Department of Medical Oncology, Dana-Farber Cancer Institute, Boston, MA, USA. [13] These authors contributed equally: Eric A Franzosa, Wendy S Garrett, Curtis Huttenhower. ✉E-mail: chuttenh@hsph.harvard.edu

members that are bioactive. We define metabolite bioactivity as its causal, responsive, or incidental association with a health, disease, or exposure-response phenotype—each of these associations have potential diagnostic, therapeutic, or (microbiome) functional characterization applications. This definition includes microbial compounds that have been found to transmit chemical signals locally (in the gut) or systemically (via circulation) in conditions such as diabetes, obesity, or the inflammatory bowel diseases (IBD) (Agus et al, 2021; Franzosa et al, 2019; Lavelle and Sokol, 2020; Lloyd-Price et al, 2019). IBD in particular is one of the best-studied complex microbiome-linked conditions, in which structural, functional, and metabolic consequences of gut dysbiosis have been identified (Franzosa et al, 2019; Lloyd-Price et al, 2019; Zhang et al, 2022). Prior stool metabolite profiling from IBD patients has demonstrated that several hundred fatty acid, secondary bile acid, triterpenoid, cholesterol, and other metabolite classes tend to be depleted during disease, while far fewer are commonly enriched (De Preter et al, 2015; Franzosa et al, 2019; Garner et al, 2007; Lloyd-Price et al, 2019). Specifically, anti-inflammatory roles have been identified for a very small number of these compounds. Short-chain fatty acids (SCFAs) are among the best-known, as enhancers of anti-inflammatory CD4+ regulatory ($T_{reg}$) cell populations via host receptor-dependent and independent pathways (Atarashi et al, 2013; Smith et al, 2013); they also influence a panoply of functions in epithelial, myeloid, and innate lymphoid cells (Chun et al, 2019; Kelly et al, 2015; Lavoie et al, 2020; Maslowski et al, 2009). Several bile acid derivatives also regulate IBD-relevant intestinal immune cells, including macrophages, dendritic cells, $T_{reg}$, and effector T cells (Thomas et al, 2022). However, a mechanistic understanding of bioactivity is currently lacking for most mass spectrometric features that are perturbed during IBD, mainly due to the lack of annotation (Franzosa et al, 2019; Viant et al, 2017).

This situation is unique neither to IBD nor to the gut microbiome overall, since features quantified in a variety of settings by chromatographic and mass spectrometry methods are frequently difficult to annotate (Chaleckis et al, 2019; Viant et al, 2017). For a small subset of features (typically a few hundred out of tens of thousands detected), annotation is possible through comparison with internal references (standards) or fully unique mass (mass-to-charge or *m/z*) matches with small molecules in databases (Franzosa et al, 2019; Lloyd-Price et al, 2019) such as the Human Metabolome Database (HMDB) (Wishart et al, 2022) and METLIN (Guijas et al, 2018). Simple mass-matching is particularly ineffective for microbial community metabolomes given its ambiguity and the underrepresentation of microbial metabolites in public databases. In addition to fully de novo chemistry, microbes often modify well-characterized standard metabolites such as amino acids, primary bile acids, and carnitine, among others into compounds that have roles in human health (Agus et al, 2021; Koeth et al, 2013; Rowland et al, 2018). Such microbial derivatives are then structurally similar to their parent standards but can have variable chemical properties due to side-chain derivatization or conformational changes. "Guilt-by-association" approaches (Edmands et al, 2017; Naake and Fernie, 2019; Uppal et al, 2017; Wang et al, 2016), which rely on the similarity of one or more physical (chromatographic retention time (RT); *m/z*) or chemical (MS2 fragmentation spectra; molecular fingerprints) properties or co-abundance between features to identify putative conjugates or derivatives, have been successfully adapted from

similar work in gene and protein function annotation (van Dam et al, 2018; Zhou et al, 2005). The Global Natural Product Social Molecular Networking (GNPS) platform is the main repository for this type of information (Wang et al, 2016), by leveraging similarity of MS2 fragmentation spectra to cluster features (including, relatedly, the identification of novel bioactive bile acid conjugates in IBD (Quinn et al, 2020)). For metabolomics datasets lacking MS2 information, correlated abundances across samples have been useful in deciphering potential functional associations among features (Franzosa et al, 2019; Naake and Fernie, 2019; Uppal et al, 2017). Covarying features are 15× more likely to belong to the same chemical class (Franzosa et al, 2019), and covariation between identified compounds and epidemiologically relevant unknown features can aid annotation of the latter. However, phenotypes such as IBD often involve perturbation of several thousand features (Franzosa et al, 2019; Lloyd-Price et al, 2019), making it challenging to simultaneously annotate and prioritize them for rigorous experimental characterization and translational applications.

We thus developed a method to prioritize potentially bioactive metabolomic features from microbial communities, using it to explore 546 single-MS (MS1) gut metabolomes from the Integrative Human Microbiome Project (iHMP or HMP2) spanning 80 IBD patients and 26 non-IBD (control) individuals that were longitudinally profiled (Lloyd-Price et al, 2019). This assigned prioritizations based on epidemiology, ecology, and molecular properties to 37,201 metabolic features, about 99% of which were unannotated. In total, 15,482 of these were newly associated with at least one standard metabolite based on covarying abundance. From these, 2672 unannotated features associated with well-characterized metabolites spanning 33 chemical classes were both highly prioritized and consistently perturbed with respect to potential bioactivity in IBD. These included metabolites placed into classes previously implicated in IBD such as bile acids (Chen et al, 2019) and short-chain fatty acids (SCFAs) (Li et al, 2021), as well as classes with lesser-known roles in gut inflammation such as bilirubins, vitamins, and polyamines. The implementation of this method (MACARRoN or Metabolome Analysis and Combined Annotation Ranks to pRioritize Novel bioactives) is available as open-source and is generalizable to any metabolomics profile or technology. Its confirmed predictions in IBD, including nicotinamide riboside and potentially novel bilirubin compounds, expand our understanding of chronic disease states and provide novel candidates for additional mechanistic characterization and drug development efforts.

## Results

### Prioritizing potentially actionable bioactive metabolites from the stool metabolome

Our identification of novel, potentially bioactive gut microbial compounds in IBD began with 81,867 unique features identified from 546 HMP2 stool metabolomes (Lloyd-Price et al, 2019), which contained an average of 50,090 (SD = 2937) features per sample. Here, a feature is a chemical entity characterized by a unique combination of RT and *m/z* (accuracy of +/− 5 ppm). These metabolomes were profiled from 265 Crohn's disease (CD), 146

ulcerative colitis (UC) and 135 non-IBD stool samples from 106 participants, followed longitudinally for up to one year each (average of 5.2 (SD = 1.2) samples per participant).

Participants with IBD ($n = 80$) contributed fewer features than non-IBD controls ($n = 26$) on average (CD ($n = 50$): 49957, UC ($n = 30$): 49511, and non-IBD: 50,910, Welch's two-sample $t$ test, CD vs. non-IBD $P$ value = 0.037 and UC vs. non-IBD $P$ value = 0.016), consistent with reduced metabolite (and microbiome) diversity in IBD (Franzosa et al, 2019; Ott et al, 2004). We classified features into three categories (Viant et al, 2017): standards, putative mass-matches, and unknowns, to profile possibly known and completely unknown features in the HMP2 stool metabolomes. Standards comprised 596 (~0.7%) features that were accurately identified using references from an internal library of 600

compounds (Fig. 1A). Putative mass-matches made up the large majority of features in each metabolome (mean = 41,305, SD = 2439), which had masses similar at least one compound in the HMDB, indicating that these features could be assigned tentative annotations but with substantial variation in expected accuracy (see "Methods"; Fig. 1A). Lastly, on an average, ~16% features in each metabolome were unknown (i.e., features that matched neither an internal reference nor any HMDB metabolite). To summarize, stool metabolomes in the HMP2 were largely uncharacterized, highlighting the importance of chemical dark matter even in well-studied environments such as the human gut and conditions such as IBD.

Next, we compared the properties of standards, putative mass-matches, and unknowns across the population. As expected,

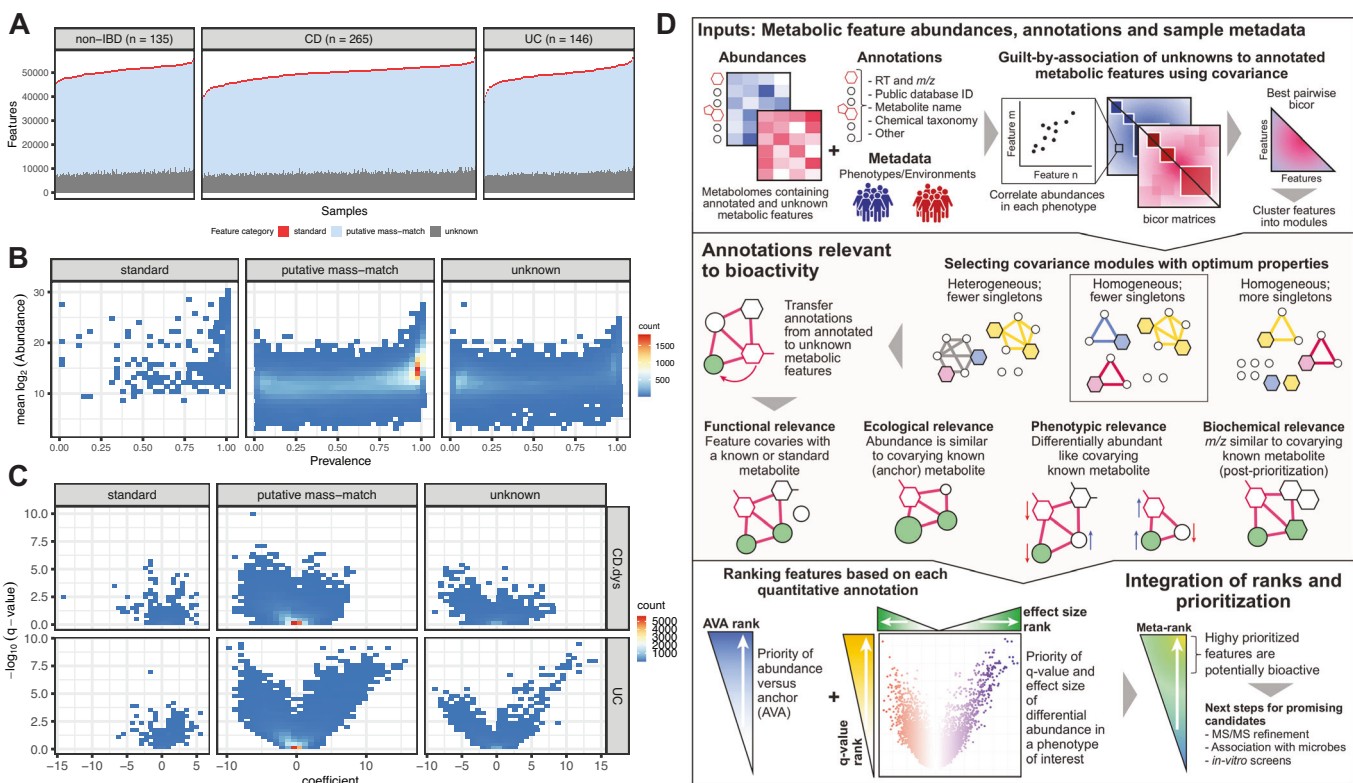

**Figure 1. Evidence of novel inflammation-linked bioactivity in the gut metabolome and methodology for prioritizing candidate compounds.**

(**A**) Overview of the annotation status of features from 546 stool metabolomes (CD: 265, UC: 146, non-IBD: 135) spanning 106 IBD patients and non-IBD controls from the Integrative Human Microbiome Project (HMP2). Each untargeted metabolome contained several thousand metabolic features which were assigned to three broad categories—standards, putative mass-match, and unknown—based on available annotations and similarity between observed $m/z$ and molecular weights of metabolites in the HMDB. Of 596 total available standards, an average of 544.3 (SD 14.8) were detected per sample (i.e., assigned an HMDB ID). A majority of features were observed to have $m/z$ similar to HMDB metabolites (putative mass-match), while many thousand could not even be assigned a sufficiently confident putative mass-match (unknowns). (**B**) Standards were enriched for high population prevalence and abundances. The abundance and prevalence of putative mass-match and unknown features were highly variable. However, several thousand were comparable to or more widespread than standards. (**C**) The effect size and $q$ value of differential abundance of prevalent metabolic features in IBD was estimated from a mixed-effects linear model by considering both disease type (CD or UC) and activity (dysbiosis) (see "Methods"). The contrasts CD-dysbiosis vs CD-nondysbiosis and UC vs non-IBD showed the greatest number of significantly perturbed features. A greater fraction of putative mass-match and unknown features were significantly enriched or depleted as compared with standards, suggesting phenotypic relevance. (**D**) Since many thousand candidate bioactive compounds are difficult to screen directly, we developed a computational method for prioritizing promising bioactives based on chemical, ecological, and phenotypic properties from uncharacterized metabolomes (MACARRoN, Metabolome Analysis and Combined Annotation Ranks to pRioritize Novel bioactives). The first step is based on the principle of "guilt-by-association," where modules based on covarying abundances are used to transfer putative annotations from annotated metabolites to co-clustered unannotated features. The next two steps leverage the ecological and epidemiological properties of metabolic features to determine their likelihood of bioactivity. Briefly, for each feature, abundance in comparison to a co-clustered standard (abundance vs. anchor (AVA)), effect size, and $q$ value of differential abundance in the phenotype of interest are determined. Next, ranks for each of these properties are integrated into a meta-rank that is ultimately used to prioritize features based on bioactivity (see "Methods").

standards, which included small molecules commonly found in the gut such as central carbon, nucleotide, and amino acid metabolites and bile acids were highly prevalent (detected in >90% metabolomes on an average) (Dataset EV1). Since physiologically relevant concentrations vary across metabolites, we observed a wide range of abundances spanning more than ten orders of magnitude (Fig. 1B). In contrast, prevalence was highly non-uniform for putative mass-match and unknown features, although many among the more prevalent features had comparable abundances to the standards (Fig. 1B; Dataset EV1).

For further analyses, we first filtered 14,634 redundant features likely to be potential adducts or fragments of metabolites (see "Methods") and from the remaining 67,233 features, we considered the subset of prevalent features i.e., features that were observed in ≥70% metabolomes of at least one of the three phenotypes (diagnoses): CD, UC, and non-IBD. Of these 37,201 non-redundant, prevalent features which included 544 standards, 12,220 (~33%) were significantly disrupted during IBD (linear mixed-effects model, $q$ value < 0.05, see "Methods"). The model includes both diagnosis (CD/UC/non-IBD) and disease activity (dysbiosis/nondysbiosis) as regressors. Comparisons between UC and non-IBD and CD-dysbiosis and CD-nondysbiosis yielded the most differentially abundant features: 8119 and 5400, respectively. In total, 200 of these were standards (Fig. 1C), including primary bile acids enriched and secondary bile acids and butyrate depleted in IBD, as expected (Dataset EV2) (Franzosa et al, 2019; Lloyd-Price et al, 2019). In addition, 10,941 putative matches and 1079 unknown features were significantly perturbed (either enriched or depleted). Taken together, the prevalence, abundance, and phenotype-associated perturbation of the 12,020 unannotated features pointed towards the richness of microbe-associated potential bioactives in the IBD gut metabolome.

Since this candidate pool was too large to simply test comprehensively, as is the case with many microbiome-associated phenotypes, we developed a new method that integrates diverse evidence to prioritize potentially bioactive metabolites from metabolomes. This approach transfers putative biological annotations to unannotated features based on "guilt-by-association", i.e., the likelihood that related compounds will covary in abundance; quantitatively evaluates ecological and phenotype or environment-associated properties for each feature; and prioritizes features as potentially bioactive in a phenotype or condition of interest (Fig. 1D). The method, MACARRoN (Metabolome Analysis and Combined Annotation Ranks to pRioritize Novel bioactives), first clusters features into covarying abundance-based modules, optimizing the sensitivity and specificity with which unannotated features are associated with at least one standard (see "Methods"). For this, pairwise biweight midcorrelation (bicor) is used as the measure of correlation of abundances. Next, for each feature, abundance versus co-clustered anchor metabolite (AVA) and association with the phenotype (effect size and $q$ value) are calculated (see "Methods"). These properties are used to determine the ecological and phenotypic/environmental relevance of each feature. MACARRoN combines ranks from each of these properties to prioritize metabolic features. As a result, features with abundances comparable or higher than the co-clustered anchor and significantly differentially abundant in the phenotype of interest are prioritized as potentially bioactive.

## Metabolite features with covarying abundances are functionally consistent and capture biochemical-relatedness

Covariance of a pair of metabolites is an indicator of functional relatedness such as co-occurrence in a biochemical pathway, co-synthesis by a microbe (or host), a common source such as diet, modification of a common parent compound, or abiotic fragmentation (Franzosa et al, 2019). To identify such associations as a means to extend annotations to the unannotated metabolic features in the HMP2 metabolomes, we used MACARRoN to cluster the 37,201 prevalent metabolic features into 355 modules based on covarying abundance (Dataset EV3). A very small number of features (1607; 4.3%) including 15 standards were not assigned to any module, referred to as singletons. Modules varied in size, ranging from 33 to 1116 features, and 67% ($N = 241$) of the modules had fewer than 100 features (Fig. 2A; Datasets EV3,and EV4). The 529 standards with module assignments were distributed across 124 (34.9%) modules (referred to as "annotated-modules" hereafter). The annotated-modules included 43% ($N = 16,010$) of all metabolic features, suggesting that nearly half of the prevalent but unidentified gut metabolites are partially characterizable based on covariance with a standard.

MACARRoN uses the co-membership of standards and unannotated metabolic features in a module to assign initial annotations (such as the chemical classes of covarying standards) to unannotated features, as well as determine their ecological relevance for prioritization. To ensure the legitimacy of assigned annotations, it is important that a module contains a biologically consistent set of metabolites. We therefore performed multiple evaluations to optimize the co-occurrence parameters and establish the biological validity of these modules. A previous study of stool metabolomes found that covarying metabolites are 15× more likely to belong to the same chemical class compared to random metabolite pairs (Franzosa et al, 2019). To compare our results with this observation, we estimated the "chemical homogeneity" of annotated-modules using information about the chemical class of the standards in the modules. The chemical homogeneity of a module was calculated as the ratio of the frequency of the most common chemical class to the number of standards in the module (see "Methods"). Chemical class information was available for standards in 110 annotated-modules, of which 57 contained two or more standards. Modules that are reasonably chemically homogeneous are ideal, as they are most likely to be analogs of biochemical pathways.

Remarkably, 26 (~46%) of annotated-modules containing 2–16 standards were ≥75% homogeneous, and an additional 9 were observed to be ≥ 60% homogeneous (Fig. 2B; Dataset EV4). Further, across all modules, standards representing the same HMDB ID were assigned to the same module in 83.33% of cases. This clustering strategy remained successful when further applied to an additional stool metabolomics dataset (Franzosa et al, 2019)—45% modules with ≥2 standards were ≥75% homogeneous and a further 7 (9.4%) were ≥60% homogeneous (Dataset EV5). Lastly, we sought to identify features representing 8 pentacyclic triterpene compounds, namely, 18 beta-glycyrrhetinic acid, alphitolic acid, asiatic acid, gypsogenin, hederagenin, rotundic acid, sumaresinolic acid, and tormentic acid (that were included as standards in another metabolomics study) in the HMP2. Upon matching the $m/$

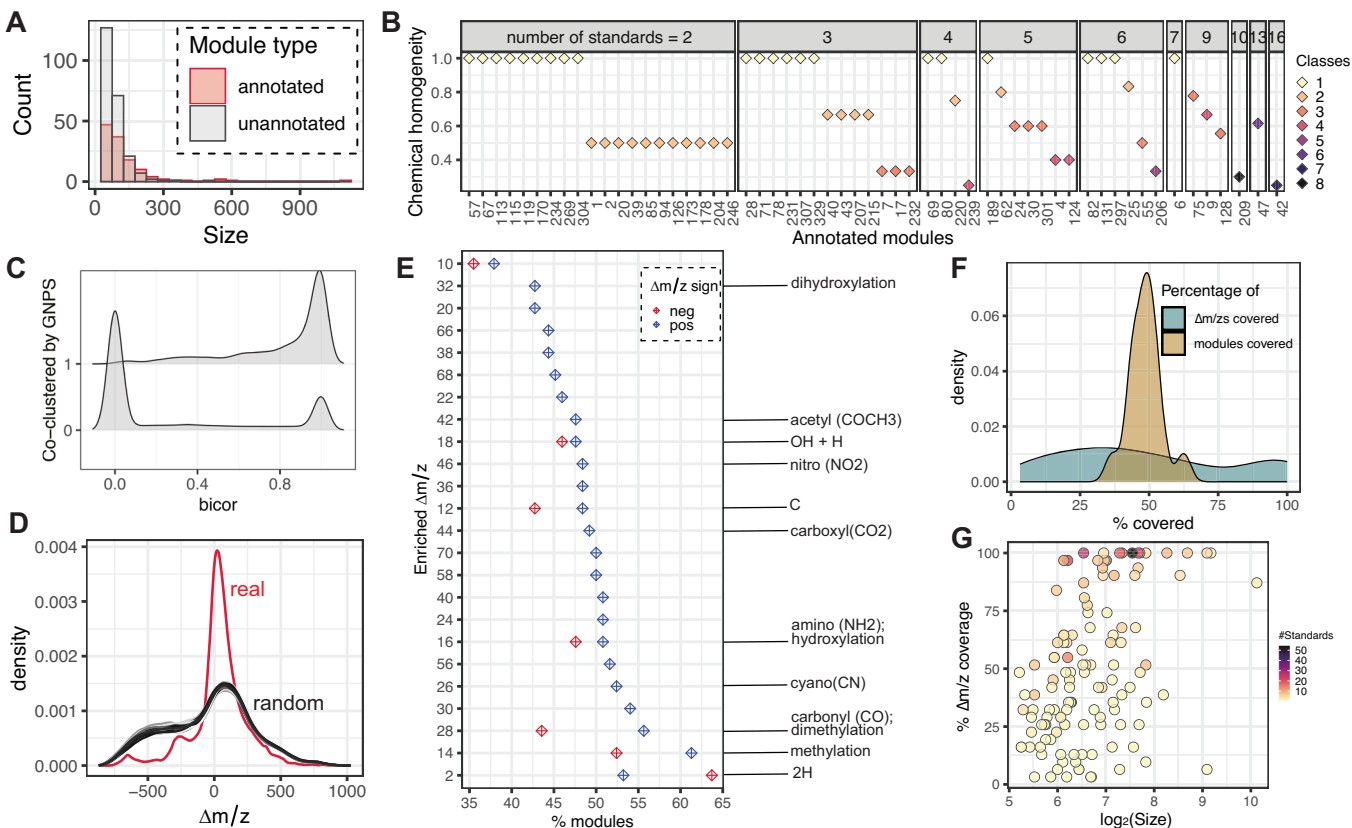

**Figure 2. Covariation in abundances produces chemically consistent modules.**

(A) 37,201 prevalent metabolic features in HMP2 metabolomes were assigned to 355 modules based on covarying abundances. In total, 124 modules that contained at least one standard compound were labeled "annotated." Both annotated and unannotated modules spanned biologically relevant sizes from 33 to 1116 features, with 67% of modules containing <100 members. (B) Chemical homogeneity of 57 annotated-modules containing ≥2 standards was determined using the chemical taxonomy (class) information of the standards (see "Methods"). Of these, 48 (~84%) were ≥50% homogeneous and 31 (45%) were ≥75% homogeneous. (C) MACARRoN covariation in abundance (using bicor (Langfelder and Horvath, 2008)) was compared with GNPS MS2 cosine similarity networks for 101 independent sputum metabolomes. Metabolites that were chemically similar (i.e., in the same GNPS molecular network) tended to have correlated abundances, whereas chemically different metabolites did not. (D) The distribution of mass-differences (Δm/zs) between co-clustered unannotated and standard metabolites in the 124 annotated-modules was significantly different than random (Kolmogorov–Smirnov test P value < 0.01; as compared to 1000 shuffles of module labels). The real distribution also contained smaller Δm/zs, indicating chemically plausible side-chain modifications among co-clustered features. (E) A permutation test revealed 24 positive and 7 negative Δm/zs significantly enriched (FDR corrected empirical P value < 0.05) in the annotated-modules. Enriched Δm/zs were non-uniformly distributed across modules, as expected if different modules capture distinct side-chain modifications characteristic of different chemical classes. However, Δm/zs associated with common biochemical transformations (indicated) were universally more prevalent. (F) Wide ranges of module coverage (percentage of total modules in which a specific Δm/z is observed) and Δm/z coverage (percentage of total Δm/zs observed in a module) were observed. Although a few modules achieved 100% Δm/z coverage, no Δm/z was observed in all modules, again consistent with class-specific side-chain enrichments. (G) Δm/z coverage appeared to be proportional to both the size of the module and the number of standards in it.

z and RT values of the triterpenes to those of the unannotated features in the HMP2, we found that they are distributed in only 2 modules—113 and 60. Interestingly, module 113 also contains two other triterpenes—maslinic acid and oleanolic acid which are among the standards in the HMP2 (Dataset EV3). This finding further established the chemical homogeneity of covariance modules.

Next, we assessed the accuracy with which covarying abundance-based modules identify biochemically related metabolites. As mentioned earlier, the GNPS molecular networking method uses the similarity between MS2 fragmentation data (cosine score) as an analogous guilt-by-association measure (coupled with other similarities such as parent mass-differences and retention times, within or across datasets) (Wang et al, 2016). Since biochemically related metabolites that differ by functional

groups typically yield similar fragments, GNPS can thus identify potential novel derivatives or conjugates. We evaluated our covarying abundance measure (bicor) (Langfelder and Horvath, 2008) against GNPS molecular networks of 101 previously published sputum metabolomes (Quinn et al, 2019). In all, 1030 pairs of co-clustered metabolites had cosine similarities above 0.7 (see "Methods"; Dataset EV6). Among these, 82.2% metabolite pairs had bicor ≥0.6 within at least one phenotype, thus also indicating strongly correlated abundances (Fig. 2C).

We continued this analysis by applying the MACARRoN module identification process to the dataset, then directly comparing GNPS clusters to MACARRoN modules using the adjusted Rand Index (ARI) (see "Methods"). We found that although the overlap between the two was small (ARI: 0.014), it was significantly higher than those obtained from randomly generated

clusters and modules ($P$ value < 0.01; one-sample $t$ test; mean ARI with random GNPS clusters: $-2.9 \times 10^{-5}$; mean ARI with random MACARRoN modules: $2.7 \times 10^{-5}$) (Fig. EV1). We expected this to be the case since MACARRoN associates metabolic features based on covariance, which can arise from several underlying processes, while GNPS relies solely on chemical similarity which is inferred from MS2 fragmentation similarity. MACARRoN modules thus tend to be complementary to GNPS clusters: functionally unrelated compounds rarely co-cluster in either method, but compounds related for different reasons are captured by one versus the other.

Finally, we evaluated mass-differences ($\Delta m/z$s) between unannotated features and co-clustered standards from the HMP2 metabolomes. We hypothesized that if the modules are indeed functionally or biochemically consistent, the distribution of $\Delta m/z$ values should be significantly non-random. We tested this using the set of annotated-modules by shuffling module labels and comparing the resulting distributions of $\Delta m/z$s to the real distribution. These were significantly different (mean Kolmogorov–Smirnov statistic over 1000 iterations: $0.23 \pm 0.02$ ; $P$ value < 0.001) (Fig. 2D). Moreover, the higher number of smaller mass-differences in the actual distribution was noteworthy, indicating that MACARRoN often associates compounds by covariation that also likely differ in only a few small functional groups.

To quantify this, we performed a permutation test to identify $\Delta m/z$s that were significantly enriched in the annotated-modules. For this, we considered the 601 positive (unannotated $m/z$ > co-clustered standard $m/z$) and 682 negative (unannotated $m/z$ < co-clustered standard $m/z$) $\Delta m/z$s that were observed at least twice. Briefly, we estimated the frequency of the 1283 $\Delta m/z$s in random modules generated by shuffling module labels (10,000 iterations) and then calculated the respective empirical $P$ values (see "Methods"). After FDR correction, 24 positive and 7 negative $\Delta m/z$s were found to be significantly enriched (Fig. 2E). Mass-differences associated with commonly observed molecular groups such as amino, acetyl, carbonyl, methyl, and nitro moieties were among the most enriched (adjusted $P$ value < 0.01), highlighting potential intra-module biochemical relationships. In agreement with the hypothesis that some chemical classes are more likely to attract certain functional groups than others, the enriched $\Delta m/z$s were found to be non-uniformly distributed across annotated-modules, with module coverage ranging from 35.5 to 63.7% (Fig. 2E,F). Notably, the most common molecular groups were also the most evenly covered, again in agreement with their ubiquity in biochemical pathways. Similarly, we observed that the coverage of $\Delta m/z$s, i.e., percentage of total $\Delta m/z$s observed in each module, varied considerably, from 3.2 to 100% (Fig. 2F). Expectedly, $\Delta m/z$ coverage was observed to be size-dependent, with larger modules having a higher $\Delta m/z$ coverage (Fig. 2G). It was also interesting to note that the instances where smaller (size <100) modules had higher $\Delta m/z$ coverage (e.g., modules 128, 209, 189) were associated with higher numbers of standards. Taken together, our analyses based on chemical homogeneity and mass-differences showed that covarying abundance modules contain metabolites that are functionally related.

## Potential bioactives covary with IBD- (or gut-) relevant standard compounds

Approximately 33% of prevalent features were significantly perturbed during IBD, including well-studied metabolite families such as bile acids and SCFAs in addition to many unidentified features (Fig. 1C). To assess the diversity and importance of these unidentified IBD-linked features, we studied the chemical diversity of standards with which they covaried, as well as their abundances relative to the same standards. In our first step to characterize the unidentified potential bioactives, we labeled annotated-modules as "IBD-relevant" if ≥25% metabolic features in them were significantly ($q$ value < 0.05) perturbed with respect to at least one of the four categories i.e., (1) CD-dysbiosis enriched, (2) UC enriched, (3) CD-dysbiosis depleted, or (4) UC depleted. Overall, 40 annotated-modules were IBD-relevant by this definition and also contained standards for which chemical taxonomy was available (Fig. 3A). Most modules were found to predominantly contain either depleted or enriched metabolic features (Fig. 3A). All modules contained features that were significantly perturbed in both CD-dysbiosis and UC, in agreement with their expected commonalities in pathogenesis and manifestation (Fig. 3A). CD-dysbiosis and UC shared seven depleted and one enriched modules, and we did not detect any modules that were enriched in one IBD subtype and depleted in the other.

In total, 43 chemical subclasses were associated with the IBD-relevant modules, spanning multiple biological processes (Fig. 3A). In agreement with previous studies of identified compounds (Franzosa et al, 2019), subclasses such as amino acids, peptides, and analogs, bile acids, and fatty acids and conjugates occurred in both enriched and depleted modules. However, a large majority of subclasses, including nucleosides, vitamins, steroids, medium-chain hydroxy acids, and carboximidic acids, were unique to one category of differential abundance. We assessed the ecological properties of these features, noting that all modules contained unidentified features that were at least 10% as abundant as the co-clustered standard (AVA ≥ 0.1) (Fig. 3B). Further, 28 (~70%) modules contained unidentified metabolic features that were more abundant (AVA > 1) than the co-clustered standard. This provides the first piece of evidence that these unidentified features may be as or more bioactive in IBD than the standard itself, integrated with others by the remainder of the analysis method.

## Highly prioritized metabolites are chemically diverse, microbiome-linked, and include both known and less-explored bioactives

Prioritization of potentially bioactive metabolites in the IBD subtypes CD-dysbiosis and UC was performed using the meta-rank obtained by integrating percentile values of ranks from AVA, $q$ value, and effect size in each subtype (see "Methods"). Thus, each of 37,201 prevalent features had two meta-ranks or "priority scores", one for each IBD subtype. In each subtype, features with priority scores in the 90th percentile (i.e., ≥ 0.731 in CD-dysbiosis and ≥0.738 in UC) and priority score ≥0.9 were considered to be highly prioritized and very highly prioritized, respectively, in that subtype (Dataset EV7). In total, 312 modules contained at least one highly prioritized (priority score ≥0.75) feature, 222 of which were common between the subtypes. These modules included 960 features highly prioritized in both subtypes (58 of these were very highly prioritized), which themselves included both enriched and depleted metabolites, e.g., bile acids, and biotin linked to anti-inflammatory pathways (Chen et al, 2019; Skupsky et al, 2020), and the IBD biomarkers -acylcarnitines and hippurate (Smith et al, 2021; Williams et al, 2010) (Fig. 4A).

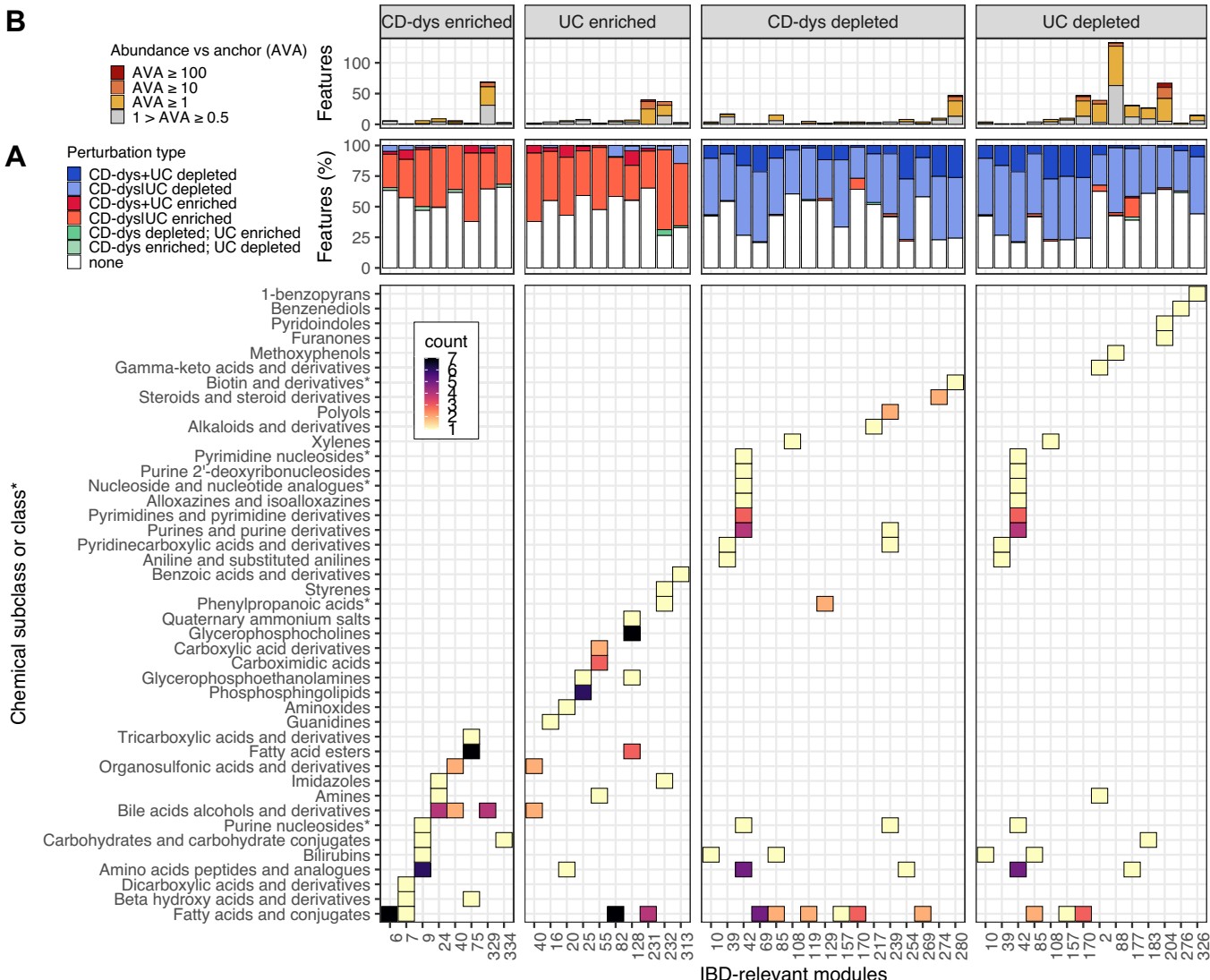

**Figure 3. Potential novel bioactives in IBD span diverse chemical classes and ecological patterns.**

(A) Top: 40 modules where ≥25% metabolic features were significantly perturbed in at least one of the four categories i.e., (1) CD-dysbiosis (CD-dys) enriched, (2) UC enriched, (3) CD-dys depleted, or (4) UC depleted, were labeled IBD-relevant. A majority of metabolic features in each module were enriched or depleted in either CD-dys or UC. Modules were categorized into four types depending on both the direction of perturbation and the disease arm of their differentially abundant metabolic features (labeled at top). Almost all modules contained >1 features similarly perturbed (enriched or depleted) in both disease arms (CD-dys + UC). Few features across different modules showed contrasting behaviors among disease arms. Bottom: Chemical taxonomy of the standards was used to determine the subclasses or classes* associated with IBD-relevant modules. A majority of subclasses/classes* were associated with a single, highly dominant direction of perturbation. (B) Abundance versus anchor (AVA) of metabolic features in IBD-relevant modules: Each module contained features that were at least 10% as abundant as the most abundant co-clustered standard (i.e., anchor metabolite). Unidentified features that appear to be as or more ecologically relevant than the standard (AVA ≥ 1) were observed in 28 modules.

Notably, a curated list of 36 standard metabolites previously published as CD associated were well-predicted by MACARRoN's prioritization (AUC: 0.893, Dataset EV8) (Franzosa et al, 2019; Gallagher et al, 2021). These CD-linked metabolites were distributed across a small subset of 15 modules, each also containing highly prioritized unannotated metabolic features that often outranked the standards themselves (Fig. 4A). Several of these unannotated features also differed from co-clustered standards by small mass-differences (median $|\Delta m/z| = 27.98$), indicating potential derivatives or conjugates. This distribution is thus supportive of the methodology, since known immunomodulatory metabolites

were included, but also suggested that novel derivatives within various metabolite classes could be equally or more important in disease. Significantly, we observed prioritization of recently identified derivatives of lithocholic acid (Paik et al, 2022) and cholic acid (Quinn et al, 2020). Isolithocholic acid, a modulator of $T_H17$ response in CD, was very highly prioritized (priority score: 0.959) and anchored by lithocholic acid in module 274. Similarly, three amino acid conjugates of cholate, anchored by cholate in module 24, were moderately- and highly prioritized in CD (priority scores—leucocholic acid: 0.746, tyrosocholic acid: 0.663, and phenylalanocholic acid: 0.789) (Fig. 4A; Dataset EV7).

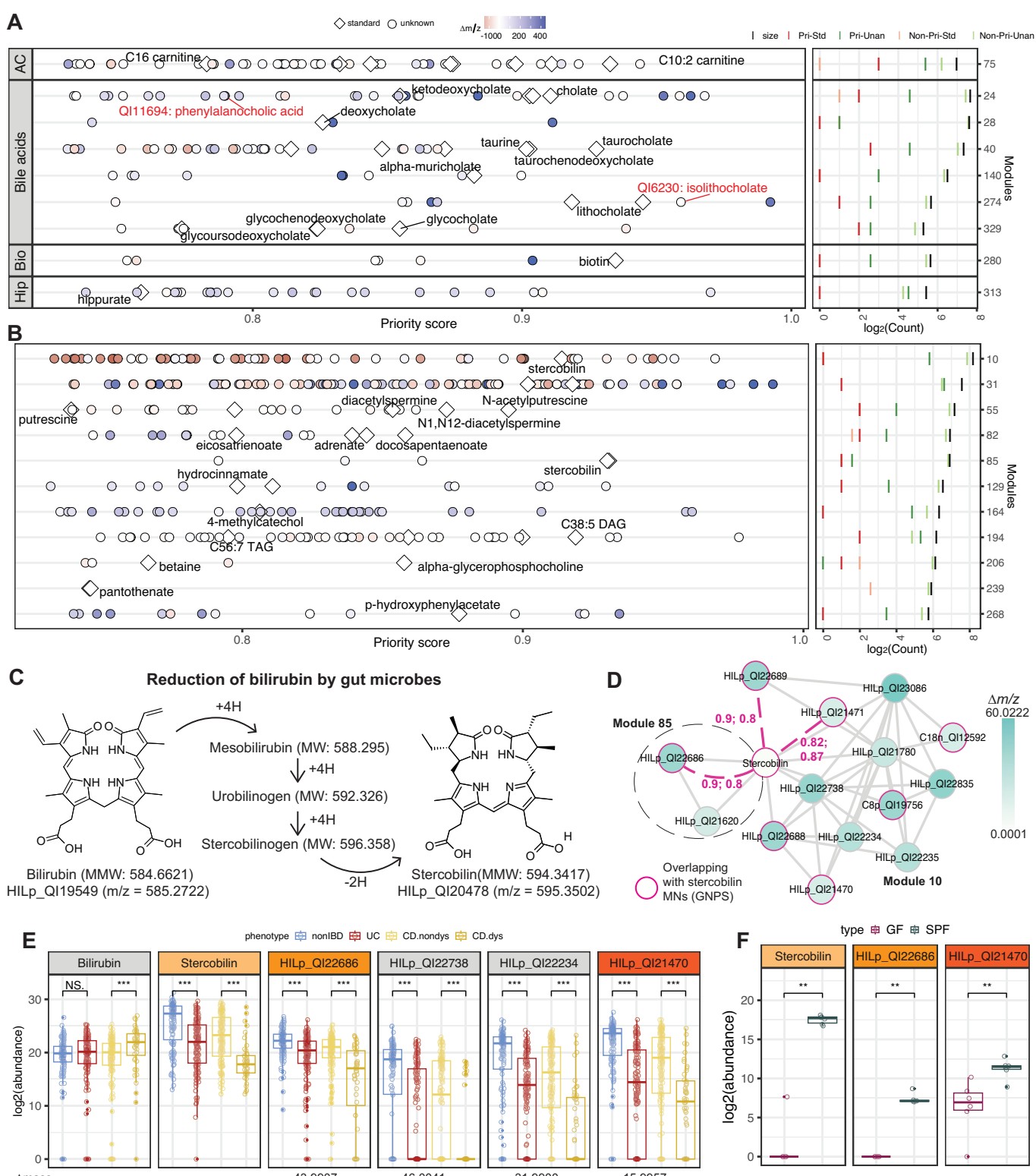

Relatedly, metabolite classes not previously linked directly to inflammation in the gut were also highly prioritized. These included vitamins, their derivatives such as pantothenate and 1-methylnicotinamide, glycerides, betaine and choline derivatives, and more interestingly, metabolites linked to gut microbial

metabolism such as hydrocinnamate (Menni et al, 2020), 4-methylcatechol (Kolomytseva et al, 2007), stercobilin (Vitek et al, 2006), N-acetylputrescine (Murray et al, 1993), and p-hydroxyphenylacetate (Saito et al, 2018) (Fig. 4B). Consistent with the loss of microbial diversity in IBD, hydrocinnamate itself,

**Figure 4. MACARRoN prioritizes known and novel bioactives in IBD.**

(A) Standards for compounds known to be IBD-linked such as IBD biomarkers and metabolites with confirmed roles in inflammation, such as acylcarnitines (AC), bile acids, hippurate (Hip.), biotin (Bio.), bilirubin (Bil.), and the short-chain fatty acid butyrate (SF), were highly prioritized (priority score >0.75) by MACARRoN. In each module containing such a known bioactive standard (total modules = 9), unannotated features were also highly prioritized ranging from 3 in module 28 anchored by deoxycholate and 49 in module 75 anchored by carnitine. The size of each module along with the number of prioritized standards (Pri-Std) and unknowns (Pri-Unk) and non-prioritized metabolic features (non-Pri-Std and non-Pri-Unk) in it are shown. Pri-Unk features typically differed from their respective co-clustered Pri-Stds by small $\Delta m/z$ values (median = 27.98), indicating potential biochemical relatedness. (B) Modules anchored by diverse chemical standards not typically associated with IBD were also highly prioritized in both CD-dysbiosis and UC, including vitamins, polyamines, and microbial metabolites such as stercobilin, hydrocinnamate, and p-hydroxyphenylacetate (only the higher rank among disease subtypes shown). Again, in several modules containing these standards, unannotated metabolic features differing by $\Delta m/z$s corresponding to common adducts were highly prioritized in both disease subtypes. (C) Stercobilin is produced via the gut microbial reduction of bilirubin in multiple steps involving intermediates such as mesobilirubin, urobilinogen, and stercobilinogen (D) Fifteen unannotated metabolites that covaried with stercobilin and differed from it by small masses were highly prioritized by MACARRoN. Seven of these were "overlapping" i.e., had masses similar to unidentified compounds linked to (with cosine score >0.7) stercobilin or urobilin in GNPS molecular networks of 7 other stool metabolomics datasets (Dataset EV9). Pink dashed edges represent potential structural similarity (cosine score >0.7) with stercobilin (Dataset EV9). (E) These highly prioritized stercobilin-linked were significantly depleted in CD-dysbiosis and UC ((Wilcoxon test, $P$ value < 0.05*; $P$ value < 0.01**; $P$ value < 0.001***; NS not significant)(four representatives shown, see others in Dataset EV9). (F) Two of the highly prioritized features and stercobilin were seen to be enriched in SPF ($N$ = 6; biological replicates) mice stool metabolomes compared to GF ($N$ = 6; biological replicates) indicating at their microbial origin (abundances in Dataset EV10; Wilcoxon test, $P$ value < 0.05*; $P$ value < 0.01**; $P$ value < 0.001***; NS not significant). For the boxplots in (E, F), the bounds of the box represent the first (Q1) and third quartile (Q3) of the data, and the line inside the box is the median. The whiskers extend to a maximum of 1.5(IQR) from the Q1 and Q3. Values beyond the whiskers are outliers. The minima and maxima are determined by the minimum and maximum observed values within 1.5 (IQR) from Q1 and Q3, respectively.

which is associated with higher microbial diversity, was significantly depleted in both subtypes (Menni et al, 2020). Similarly, stercobilin, the end product of microbial bilirubin metabolism, was depleted (Fahmy et al, 1972; Vitek et al, 2006). Many species of clostridia and Coriobacteriaceae can convert tyrosine to p-hydroxyphenylacetate and ultimately p-cresol (Elsden et al, 1976; Saito et al, 2018), which can be further hydroxylated to 4-methylcatechol, although primarily by species not typically associated with the gut such as *R. opacus and P. putida* (Hopper and Taylor, 1975; Kolomytseva et al, 2007). While p-hydroxyphenylacetate was enriched in both disease subtypes, 4-methylcatechol was depleted. The prioritization of p-hydroxyphenylacetate in particular is in agreement with the increased transcription of two clostridia—*C. hathewayi* and *C. bolteae* - in HMP2 metatranscriptomes (Lloyd-Price et al, 2019), as well as the previously reported expansion of Coriobacteriaceae in IBD (Alam et al, 2020).

The bilirubin family (bilirubin is represented by feature HILp_QI19549; $m/z = 585.2722$ [M + H]+) is among the most abundant in the gut. Bilirubin was highly prioritized only in CD-dysbiosis and is well-known for being metabolized in several steps to stercobilin and urobilin by the intestinal microbiota (Vitek et al, 2006). As in the case of microbial conversion of primary to secondary bile acids, the depletion of stercobilin concurrent with the enrichment of bilirubin likely indicates the loss of microbial enzymes which catalyze the reduction. Despite the centrality of microbial metabolism to bilirubin homeostasis (Hamoud et al, 2018), only a handful of clostridia, *Bacteroides fragilis*, and *Citrobacter youngae* have been experimentally shown to reduce bilirubin, and knowledge of microbe-made bilirubin derivatives is mostly limited to its approximately a dozen reduction products (Shiels et al, 2019; Vitek et al, 2006) (Fig. 4C). Bilirubin itself ameliorates colonic injury and inflammation in dextran sodium sulfate (DSS) induced murine model of inflammatory colitis via its antioxidant i.e., reactive oxygen species scavenging properties (Zheng et al, 2019; Zucker et al, 2015). While bilirubin can directly prevent leukocyte migration into the colon (Vogel and Zucker, 2016), its implication in disease and reliance on gut microbes for subsequent metabolism and eliminations suggests that its

derivatives may have additional functions (e.g., bile acids (Funabashi et al, 2020; Quinn et al, 2020)). Thus, we were particularly interested in the highly prioritized (in both CD-dysbiosis and UC) unannotated features in which were anchored by stercobilin in modules 10 and 85; again, these generally differed by small positive mass-differences typically suggestive of novel backbone derivatizations (Fig. 4B).

The 15 highly prioritized unannotated features in the modules 10 and 85 with small positive $\Delta m/z$s compared to stercobilin were also depleted (Fig. 4D,E; Dataset EV9). Upon closer examination of the differences in neutral masses of these features and stercobilin, we found five similar to the masses of oxygen (~15.996) and di-oxygen (~31.99) moieties (Dataset EV9). The other mass-differences that cannot be easily associated with small molecular groups may be the combined result of two or more reactions. To provide further data establishing the chemical similarity of these features to stercobilin, we examined the masses of compounds that were directly connected to stercobilin (cosine score ≥0.7 and $|\Delta m/z|$ w.r.t stercobilin ≤100) in the GNPS molecular networks (MNs) of 11 MassIVE human stool metabolomics datasets (Dataset EV9). Since urobilin and stercobilin are structurally very similar, we also examined the masses of compounds directly connected to urobilin when it was included in the same molecular network as stercobilin. Although we did not find exact matches between the masses of compounds in stercobilin MNs and the highly prioritized features in our dataset, in seven instances, they differed only by ~1 (i.e., neutral mass of prioritized feature—parent mass of GNPS compound ~1) (Dataset EV9; Fig. EV2). This is within the range of typical inter-dataset differences in the observed $m/z$s of stercobilin itself (593.76 (MSV000079777), 594.761 (MSV000082262), 595.349 (MSV000079651), and 595.597 (MSV000082629)), resulting in a $\Delta m/z$ of 1.832 between the largest and smallest masses. These seven compounds in the MassIVE datasets that "overlapped" with our highly prioritized features had cosine scores ≥0.8 with stercobilin or urobilin, and four differed from them by $CH_2O_2$ and $CO2$ moieties (Dataset EV9), again supporting the presence of uncharacterized disease-linked bilirubin compounds in stool.

Since stercobilin is microbially derived, we wanted to confirm if the depletion—and by extension, prioritization —of the 15 features

in modules 10 and 85 was due to perturbed microbial activity in IBD. Toward this, we profiled the abundances of these features in the fecal metabolomes of germ-free (GF) as compared with specified pathogen-free (SPF) mice using the HILIC-positive LC–MS method (11 of the 15 prioritized features were initially quantified by the same method; Datasets EV9 and EV10, "Methods"). Six features derived from human IBD profiles—including stercobilin—could be mapped to features detected in the GF and SPF mouse stool metabolomes. As expected, stercobilin was very significantly enriched in SPF mice relative to GF (Fig. 4F). More interestingly, two additional features [HILp_QI22686 ($m/z = 639.3403$ [M + H]+) and HILp_QI21470 ($m/z = 611.3453$ [M + H]+) were also significantly enriched in SPF mice (Fig. 4F), supporting their microbial origin and thus the ability of MACARRoN to prioritize novel candidate bioactives that are both phenotypically linked and microbiome-derived.

## Nicotinamide riboside validated to ameliorate dextran sodium sulfate (DSS)-induced colitis

Pantothenate (vitamin $B_5$) in module 239 was significantly depleted and highly prioritized in both CD-dysbiosis (priority score = 0.7459) and UC (priority score = 0.7455) (Fig. 4B). The other B vitamin in the same module, nicotinate (vitamin $B_3$), was also significantly depleted in both disease subtypes but moderately-prioritized (in the 75th percentile of priority scores) only in UC (Fig. 5A). In addition to these two vitamin standards, several metabolic features in module 239 were also significantly depleted in CD-dysbiosis and UC (Figs. 3A and 5A), suggesting perturbation of vitamin metabolism during dysbiosis. Nicotinate is a precursor in the gut microbiome mediated synthesis of nicotinamide adenine dinucleotide ($NAD^+$) via the salvage pathway (Shats et al, 2020) (Fig. 5B). Recent studies show that intestinal nicotinate and $NAD^+$ levels can be boosted with oral nicotinamide riboside (NR); this process requires gut microbial deamidation (via nicotinamidase $pncA$) of nicotinamide (from NR) to nicotinate and assimilation of the nicotinate moiety into $NAD^+$ (Chellappa et al, 2022). An unannotated feature in the same module—HILp_QI6481 was identified to be NR by matching its $m/z$, RT, and fragmentation with the NR standard (Fig. EV3). This observed covariance of NR with nicotinate confirms the biochemical coupling between them in the gut and also, once again, underscores MACARRoN's ability to identify substrate–product relationships from covariance.

We first inspected the gene abundances (specifically, effect sizes for dysbiosis versus nondysbiosis) of microbial enzymes catalyzing de novo (Gazzaniga et al, 2009; Kurnasov et al, 2003) and NR-salvage (Bieganowski and Brenner, 2004; Shats et al, 2020) based synthesis of $NAD^+$. Enzymes involved in both synthesis routes were depleted during dysbiosis (Fig. 5C). Here, the depletion of nicotinamidase genes ($pncA$) from several species is particularly notable since it explains the depletion of their product—nicotinate, during dysbiosis and thereby, its prioritization by MACARRoN. Taken together, these observations pointed towards decreased gut microbiome-dependent synthesis of $NAD^+$ in IBD. The salvage pathway found in the host recruits the highly conserved NR kinase (NRK) enzymes that phosphorylate NR to nicotinamide mono-nucleotide (NMN), which is then converted to $NAD^+$ by NMNAT2 (Bieganowski and Brenner, 2004). *NRK* transcription remained unchanged in IBD (Fig. 5D), however, NR salvage via *NRK* is insufficient to increase the intestinal $NAD^+$ levels (Chellappa et al,

2022). Instead, host cells also rely on microbially provided nicotinate to synthesize $NAD^+$ (Chellappa et al, 2022).

In addition to its role as a coenzyme, $NAD^+$ plays an important role in the regulation of inflammatory and immune processes (Rajman et al, 2018). Therapeutic benefit from the supplementation of $NAD^+$ precursors such as NR and NMN has been observed in inflammation-related and gastrointestinal disorders such as obesity (Canto et al, 2012), diabetes (Yoshino et al, 2011), liver fibrosis (Pham et al, 2019), and age-related colonic dysmotility (Zhu et al, 2017). NR supplementation not only replenishes the $NAD^+$ metabolome but also reduces levels of circulating inflammatory cytokines (Elhassan et al, 2019). Given these previous findings, the covariance between NR and nicotinate, and MACARRoN's prioritization of nicotinate in IBD, we asked whether NR itself had bioactivity in IBD using two preclinical models of intestinal injury and inflammation, acute and chronic DSS. In the acute model (Fig. 5E), we observed a modest and statistically significant improvement in histological colitis scoring for NR (1000 mg/kg) as compared with phosphate-buffered saline (PBS) administered intraperitoneally (two-tailed Mann–Whitney $P$ value = 0.014, Fig. 5F). In the chronic model (Fig. 5G), which endeavors to capture the relapsing and remitting nature of IBD via repeat exposure to DSS, the difference in colitis scores between the PBS and NR groups was again significant and even more pronounced (two-tailed Mann–Whitney $P$ value = 0.007, Fig. 5H). Collectively, these results validate both the prioritization of nicotinate and bioactivity of NR in IBD and, additionally, demonstrate MACARRoN's utility to locate potential bioactives (NR) in large datasets by virtue of their covariance with prioritized, well-characterized metabolites (nicotinate).

## Discussion

Although untargeted metabolomics is routinely used to profile the chemical repertoires of microbial communities, most chemical features identified by such methods are difficult or impossible to identify, requiring substantial effort to prioritize and validate. This is true even for metabolomes of well-studied environments, such as the human gut, where the microbiome catabolizes or produces health-relevant metabolites. Here, we present a framework for prioritizing the most promising and actionable metabolites from among thousands of features in untargeted metabolomes associated with phenotypes of interest. We applied this method (MACARRoN) to ~67k metabolic features derived from IBD and control populations, finding associations indicating potential biochemical or functional relationships between ~500 known, gut-relevant metabolites and 15,481 unannotated metabolic features. Prioritization highlighted both well-characterized—e.g., bile acid and SCFA derivatives—and modestly explored classes of metabolites such as polyamines, bilirubins, and vitamins. Altogether, this provides information to initially characterize nearly 23% of the gut metabolome. Finally, we annotated and validated anti-inflammatory activity of nicotinamide riboside in IBD, providing an end-to-end demonstration of MACARRoN's ability to find compelling candidates for downstream characterization from complex untargeted metabolomes.

Data integration for microbial community metabolomics can thus provide both candidate bioactive leads in phenotypes

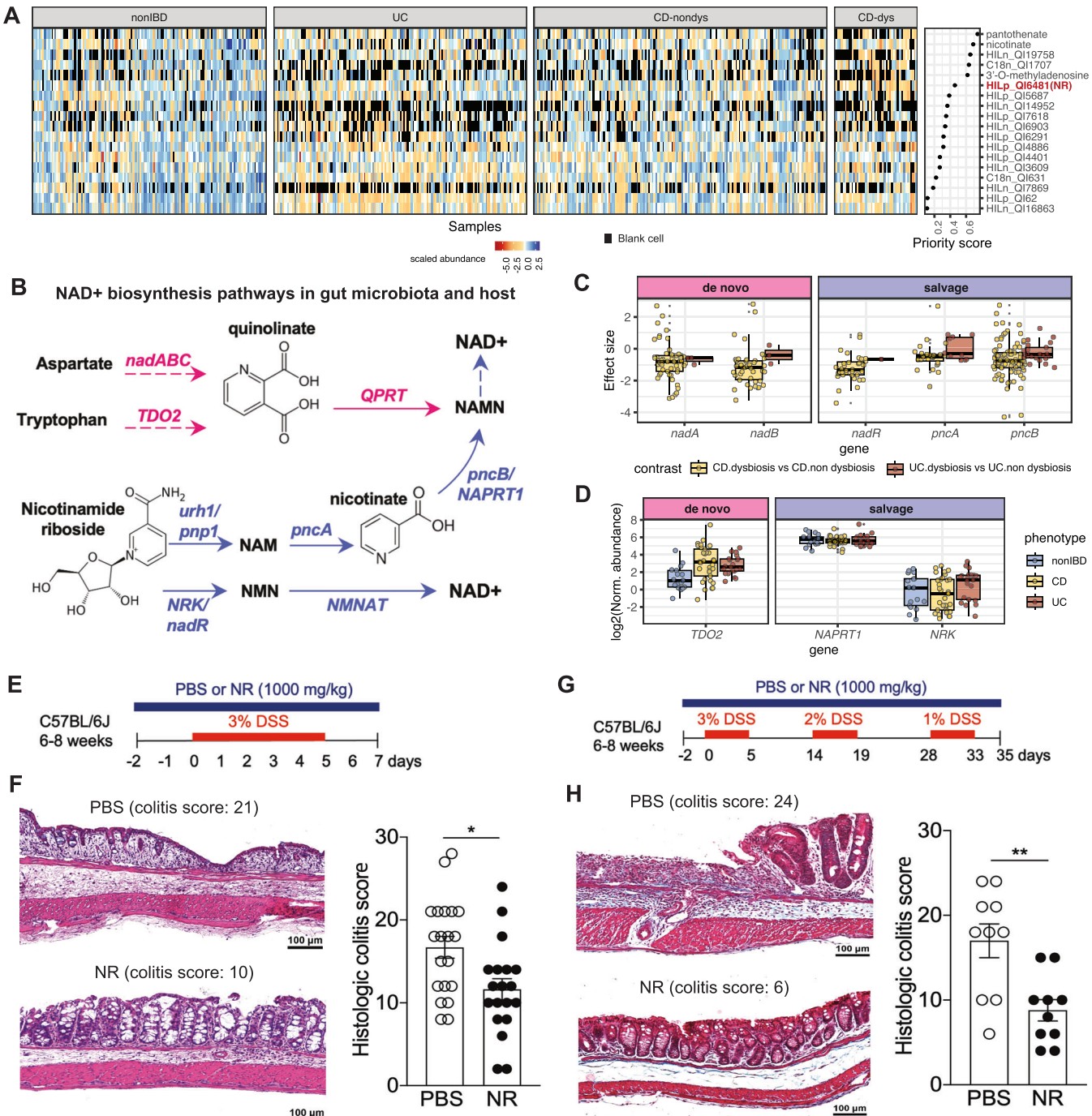

such as inflammation, as well as a foundation for more basic characterization of the gut (and other environments') metabolome. In this study's data, prioritized (priority score ≥0.6) metabolites covaried with standards belonging to 43 chemical subclasses (structure-based chemical taxonomy defined by ChemOnt (Djoumbou-Feunang et al, 2016)). This shows, first, the unappreciated diversity of IBD-linked metabolic pathways, since previous studies have focused on small subsets of validated standards such as bile acids (Chen et al, 2019), butyrate (Chen et al, 2018), bilirubin (Li et al, 2021), and tryptophan

derivatives (Scott et al, 2020). Second, relatedly, the combination of disease (or other phenotype) epidemiology with co-clustered standards and chemical properties allows us to locate potentially bioactive unannotated metabolic features even in the absence of MS-MS or other additional experimental profiles. This is demonstrated by the prioritization of the recently identified lithocholate and cholate derivatives, microbially associated metabolites that covary with stercobilin, and by the experimentally validated bioactivity of nicotinamide riboside in IBD (Figs. 4A,F and 5A,F,H).

**Figure 5.  Validation of nicotinate and NR as IBD-linked bioactives.**

(A) Abundances of metabolic features in nicotinate/pantothenate-anchored module 239 that were depleted (*q* value < 0.1 and |effect size| ≥ 1) in CD-dysbiosis or UC and nicotinamide riboside (NR) represented by HILp_QI6481. Priority is indicated in terms of the best-observed priority score between the two disease subtypes. (B) NAD is synthesized from Asp and Trp via the de novo pathway or from nicotinamide riboside (NR) via the salvage pathway. Microbial and host enzymes that catalyze de novo (shown in pink) and salvage reactions are indicated. (C) Microbial enzymes catalyzing both de novo and salvage pathways were depleted in gut metagenomes during active disease i.e., IBD (particularly CD) dysbiosis (CD.dysbiosis, *N* = 177; CD.nondysbiosis: 555; UC.dysbiosis: 51; UC.nondysbiosis: 386). (D) Transcription (log$_2$(RPKM)) of the host enzymes catalyzing de novo (*TDO2*) and salvage (*NRK*) pathways was enriched and unchanged, respectively (CD, *N* = 33; UC: 21; non-IBD: 17). For the boxplots in (C, D), the bounds represent the first (Q1) and third quartile (Q3) of the data, and the line inside the box is the median. The whiskers extend to a maximum of 1.5 (IQR) from the Q1 and Q3. Values beyond the whiskers are outliers. The minima and maxima are determined by the minimum and maximum observed values within 1.5 (IQR) from Q1 and Q3, respectively. (E) Acute DSS model used 3% daily exposure in drinking water for 5 days per mouse, and compared groups that received intraperitoneal (IP) NR vs. PBS (control). (F) Images from hematoxylin and eosin-stained formalin-fixed paraffin embedded (FFPE) slides from each group (acute DSS model), paired with histologic colitis scores (*N* = 20 mice per group). (G) Chronic DSS model, using decreasing DSS percentages across three 5-day windows, again compared NR vs. PBS control. (H) Images from Masson's trichrome (used to highlight the chronic model's characteristic fibrosis) stained FFPE slides from each group, (chronic colitis model) again paired with histologic colitis scores (*N* = 10 mice per group). Data in (F, H) represent two independent experiments and each symbol corresponds to data from an individual mouse. (F, H) Mean + SEM shown, two-tailed Mann–Whitney *U* test, *P* value < 0.05*, *P* value < 0.01**. Source data are available online for this figure.

Other prioritized chemical classes corresponded with broad changes in the gut microbiome concordant with dysbiosis, which may be reflective of broader consequences of host–microbe interactions. These included depletion of metabolic indicators of microbial diversity such as hippurate and hydrocinnamate, along with the enrichment of p-hydroxyphenylacetate, a product of tyrosine metabolism by clostridial and Coriobacteriaceae species that are themselves enriched in IBD (Alam et al, 2020; Lloyd-Price et al, 2019; Saito et al, 2018). Other examples of prioritized metabolite classes that also corresponded with overall microbiome disruption included pathways such as bilirubin, polyamine, and vitamin metabolism that lie at the interface of host-microbial interactions (Figs. 4B,C and 5A). Remarkably, stercobilin, polyamines, and several unannotated features were ranked even higher by MACARRoN than well-studied IBD bioactives like bile acids, bilirubin itself, and butyrate, alluding to the richness of unknown but potentially critical metabolites during inflammation.

Of these, members of the bilirubin/stercobilin pathway may be of particular interest. Both bilirubin and urobilinogen are antioxidants (Vogel and Zucker, 2016) and additionally, bilirubin has been shown to be anti-inflammatory (Zucker et al, 2015) and influence bacterial survival in the gut (Nobles et al, 2013). Both of these properties are of therapeutic interest in IBD management (Vitek and Tiribelli, 2020), motivating us to examine the 15 highly prioritized unknown features that covaried with stercobilin. We found compounds in the GNPS stercobilin molecular networks that were similar in mass to seven features that were highly prioritized by MACARRoN, supporting the chemical relatedness of the latter to bilirubins. The mass-differences associated with these features suggested small molecular additions to the tetrapyrrole backbone much like in cases of microbe-made amino acid conjugates of bile acids (Quinn et al, 2020). Notably, a recent study reported the intra-duodenal conversion of biliverdin to bilirubin-10-sulfonate by *C. youngae* (Shiels et al, 2019), indicating the possibility of other microbial bilirubin derivatives. Our mouse stool metabolomics data supported that two prioritized compounds were of microbial origin like stercobilin itself (Fig. 4F). Despite the small overlap between the number of prioritized and validated bilirubin-associated compounds (previous studies (Han et al, 2021; Marcobal et al, 2013; Meier et al, 2023) may be used to infer the broader overlap between (IBD-associated) human and SPF-specific metabolites belonging to other chemical classes), the findings are still notable

given many factors that differentiate this system from human IBD: lower gut microbial diversity in SPF mice, markedly different diets as compared to humans, absence of inflammation as in dysbiotic human subjects, and thus, potential absence of host-derived or inflammation-linked precursor substrates. Our results here thus provide a new incentive to further characterize bilirubin pathway members in the gut. Relatedly, validation of the therapeutic benefit from NR supplementation in a mouse model of colitis is an encouragement to further study the prioritized bioactives for therapeutic bioprospecting.

MACARRoN's strategy for compound prioritization and identification is fundamentally different from that of existing guilt-by-association methods such as GNPS (Wang et al, 2016), xMSAnnotator (Uppal et al, 2017), and MetNet (Naake and Fernie, 2019) that are also designed for chemically annotating features. Each method uses some combination of properties including *m/z*, RT, fragmentation spectra, and covariance to associate and annotate metabolic features. However, MACARRoN is uniquely able to leverage microbial community profiles, as well as the intuition that covariation captures molecular functional relatedness as previously used for protein function prediction (Zhou et al, 2005). It also combines these and other more basic properties such as prevalence and abundance with respect to a known metabolite with phenotypic "importance" in prioritization of potential bioactivity, i.e., the likelihood of its causal or consequential involvement in a phenotype or an environment of interest. This simultaneously indicates which of potentially thousands of unidentified compounds may be most important to experimentally validate, and what putative identities they might be validated to have. Although MACARRoN does not use properties such as the *m/z* or RT directly, these can still be used post hoc to make informed guesses about the identity of an unannotated metabolite that covaries with a known metabolite.

This combination of methodology provides MACARRoN with several advantages relative to other approaches. Unlike annotation methods that associate metabolic features using covariance and filter associations based on chemical properties (e.g., xMSAnnotator (Uppal et al, 2017), MetNet (Naake and Fernie, 2019)), MACARRoN uses only covariance. This can be either a strength or a weakness—it loses the specificity provided by spectral similarity (when available), but it does not require this additional data to operate and can thus prioritize bioactive compounds that are less

directly related to characterized precursors or derivatives. Apropos, it is notable in these results that despite the absence of chemical property-based filtering, most covariance-based modules are chemically homogeneous (Fig. 2B). Moreover, metabolites from the same GNPS molecular networks were also found to have correlated abundances by MACARRoN (Fig. 2C), although the latter does not rely on spectral information. MACARRoN can thus analyze low molecular weight metabolites that do not have complex spectra even when MS2 profiles are generated. This makes the method particularly suitable for identifying microbially processed small molecules such as amino acids, polyamines, SCFAs, and other derivatives of common molecular classes. In this regard, it is important to note that the prioritization schema itself is agnostic to the source or origin of metabolite i.e., MACARRoN will prioritize both microbial and non-microbial metabolites, however, in microbiome-linked phenotypes such as IBD, the prioritized metabolic features are expected to be enriched for microbially derived small molecules. Finally, MACARRoN does not require curated biochemical pathway information such as KEGG (Kanehisa and Goto, 2000) or MetaCyc (Caspi et al, 2016), making it appropriate for less well-studied environments such as microbial communities. Finally, to the best of our knowledge, MACARRoN is the only tool that couples characterization with prioritization based on the likelihood of bioactivity, i.e., chemical involvement in a phenotype of interest. This step is instrumental for efficiently selecting the most promising metabolites for further characterization.

Conversely, MACARRoN of course also has limitations, the most striking being the nearly 50% of highly prioritized metabolites not associated with any standard, rendering their characterization nearly impossible by these methods. Relatedly, chemically identifying metabolites using only chemical properties and mass-differences with respect to co-clustered standards is nontrivial. The uncertainty in identifying a metabolite arises from multiple underlying phenomena such as abiotic fragmentation and adduct formation in the mass-analyzer, transformations that involve the simultaneous removal and addition of moieties, and ambiguity between $\Delta m/z$s (e.g., $\Delta m/z$s near 44 are associated with both carboxyl and amido groups). It may thus be potentially useful to couple the system with tools such as BioTransformer (Djoumbou-Feunang et al, 2019) and MetWork (Fox Ramos et al, 2019) that predict metabolic products obtained from the host or microbial metabolism of a parent compound. MACARRoN can also be used in tandem with related systems for MS1 metabolomes such as xMSAnnotator (Uppal et al, 2017) and MetNet (Naake and Fernie, 2019) for the chemical identification of prioritized metabolites. Additional limitations arise in the optimization of MACARRoN's methodological parameters including the requirement of sufficient samples per phenotype for detecting significant correlations; accompanied by challenges routinely associated with clustering such as optimizing cluster size and membership. While we expect the defaults to work for most real-world metabolomics datasets, to determine if the modules generated using default parameters are appropriate for downstream prioritization steps, we provide the users descriptive statistics pertaining to the number of modules, singletons, and chemical homogeneity of the modules, as well as the option to adjust the defaults. We also recognize that different metabolites are bioactive at different concentrations and therefore, determine the ecological relevance of a metabolite by comparing its abundance to that of a co-clustered standard i.e., AVA. However, in larger modules where abundances of metabolites span several orders of magnitude, there is a risk of losing metabolites that are bioactive at low concentrations. Finally, for microbial community-associated metabolomes, the current implementation cannot distinguish between host and microbial metabolites which could be incorporated in the future by leveraging associations with features in paired metagenomic datasets.

Although untargeted metabolomics is a powerful resource for the quantification of the metabolic activity in a phenotype or environment of interest, options for interpreting the many thousands of resulting, unidentified metabolic features remain limited. We developed MACARRoN to bridge the gap between untargeted metabolomics and downstream end goals such as bioprospecting, biomarker identification, and compound and enzyme characterization. All of these are particularly relevant in —although not unique to—the under-explored environment of microbial communities and the human microbiome. By integrating functional association, ecological, and phenotype-related information of metabolic features into a prioritization scheme, MACARRoN provides initial annotations for phenotype-linked metabolite features while simultaneously ranking them based on their potential bioactivity. This reduces the search space for actionable metabolites by several orders of magnitude. Using this approach, we identified several new metabolite classes and derivatives implicated in pro- or anti-inflammatory activity during IBD. We hope that future applications of the method will be able to illuminate the chemical dark matter which currently limits our understanding of complex phenotypes and environments.

# Methods

Reagents and tools table.

| Reagent/ resource | Reference or source | Identifier or catalog number |
|---|---|---|
| **Datasets analyzed** | | |
| HMP2 metabolomics | HMBR data portal (https://portal.microbiome-bioactives.org/) | https://downloads.microbiome-bioactives.org/wgs/HMP2/data/wgs/metabolites/HMP2_metabolites_all_methods.tsv |
| HMP2 host transcriptomics | IBDMDB (https://www.ibdmdb.org/) (Lloyd-Price et al, 2019) | https://www.ibdmdb.org/downloads/products/HMP2/HTX/host_tx_counts.tsv.gz |
| Cystic fibrosis sputum metabolomics | (Quinn et al, 2019) | |
| **Experimental models** | | |
| Germ free C57BL/6J (*M. musculus*) | Bred at the Harvard T. H. Chan School of Public Health | |
| SPF C57BL/6J (*M. musculus*) | Jackson Laboratory | ISMR_JAX:000664 |
| **Chemicals, enzymes, and other reagents** | | |
| Dextran sodium sulfate (DSS) | Thermo Fisher Scientific | J1448922 |

| Reagent/ resource | Reference or source | Identifier or catalog number |
|---|---|---|
| Nicotinamide riboside (NR) | PCI Synthesis, Newburyport, MA | 23111-00-4 |
| PFA | Sigma-Aldrich | 441244 |
| **Databases, software, and libraries for bioinformatic and statistical analyses** | | |
| GNPS | https://gnps.ucsd.edu/ (Wang et al, 2016) | |
| MetaWIBELE v0.3.8 | https://github.com/ biobakery/metawibele (Zhang et al, 2022) | |
| WGCNA v1.72-1 | https://cran.r-project.org/web/ packages/WGCNA/ index.html (Langfelder and Horvath, 2008) | |
| MaAsLin2 v1.12.0 | https://www. bioconductor.org/ packages/Maaslin2/ (Mallick et al, 2021) | |
| limma v3.54.2 | https://www. bioconductor.org/ packages/limma/ (Ritchie et al, 2015) | |
| tidyverse v2.0.0 | https://cran.r-project.org/web/ packages/tidyverse/ index.html | |
| data.table v1.14.8 | https://cran.r-project.org/web/ packages/data.table/ index.html | |
| mclust v6.0.0 | https://cran.r-project.org/web/ packages/mclust/ index.html (Scrucca et al, 2016) | |
| HMDB v5.0 | https://hmdb.ca/ (Wishart et al, 2022) | |
| IBDMDB | https://www. ibdmdb.org/ (Lloyd-Price et al, 2019) | |
| HMBR data portal | https://portal. microbiome-bioactives.org/ | |
| **Libraries for visualizations** | | |
| ggplot2 v3.4.1 | https://cran.r-project.org/web/ packages/ggplot2/ index.html | |
| ggrepel v0.9.3 | https://cran.r-project.org/web/ packages/ggrepel/ index.html | |
| ggpubr v0.6.0 | https://cran.r-project.org/web/ packages/ggpubr/ index.html | |
| cowplot v1.1.1 | https://cran.r-project.org/web/ packages/cowplot/ index.html | |

| Reagent/ resource | Reference or source | Identifier or catalog number |
|---|---|---|
| RcolorBrewer v1.1-3 | https://cran.r-project.org/web/ packages/ RColorBrewer/ index.html | |

## Methods and protocols

### The MACARRoN algorithm for putative bioactive prioritization

MACARRoN is a set of methodologies for prioritization and initial annotation of small molecules with potential bioactivity in phenotypes of interest from large-scale metabolomes. Untargeted metabolomics provide abundances for typically several thousand metabolic features per environment, only very few of which can be confidently identified (i.e., assigned annotations by chemical standards). To accommodate typical MS-based untargeted chemical profiles, MACARRoN requires as inputs; (1) MS1 abundances of metabolic features, (2) systematic identifiers (HMDB (Wishart et al, 2022) or PubChem (Kim et al, 2021)) for a priori identified metabolite standards, and (3) per-sample metadata including host or environment phenotypes of interest. The workflow employs a three-step strategy to associate, quantitatively annotate, and prioritize metabolic features. MACARRoN (https://huttenhower.sph.harvard.edu/macarron/) is available as an R package from Bioconductor (https://bioconductor.org/packages/Macarron) and can also be executed via the command line.

**Guilt by association**. In its first step, MACARRoN clusters untargeted metabolic features into covarying modules based on similar abundance profiles. To do this, first, metabolic features that are prevalent i.e., observed in at least a user-defined fraction of samples of at least one phenotype are selected (default 70%). Next, for each pair of prevalent metabolic features, biweight midcorrelation (bicor) (Langfelder and Horvath, 2008) is calculated in a stratified manner considering samples from each phenotype at a time. This is done to avoid false positive correlations arising due to differences in abundances in two conditions. Correlations between metabolites are systemic properties. Noise in complex biological systems such as the gut makes interpretation of underlying biological phenomena from metabolite correlations challenging (Steuer, 2006). This is especially true for weak to moderate negative correlations, which may arise due to complex pathway structure, enzyme abundances, and pleiotropy as well as system-level perturbations such as diet and antibiotics. Therefore, MACARRoN considers only positive correlations and negative bicor values are replaced with 0.

As recommended by previous studies for scale-free topology, the best-observed correlation is cubed and used for tree construction with average-linkage hierarchical clustering (Langfelder and Horvath, 2008; Mock et al, 2018). For the detection of modules (i.e., clusters of highly correlated metabolites), MACARRoN implements the dynamic-hybrid algorithm to build clusters in a "bottom-up" manner (Langfelder et al, 2008). Briefly, modules that contain a certain minimum number of metabolic features ("minClusterSize") are detected and then smaller modules and unassigned metabolic features (singletons) are progressively merged into larger modules

that satisfy a predefined criteria for being distinct. MACARRoN uses the cube root of the total number of prevalent features as the "minClusterSize". This choice of minClusterSize yields functionally homogeneous clusters and fewer singletons and is expected to be applicable to most real-world metabolomic datasets ("Results"). The resulting modules are either annotated (contain standards) or unannotated.

**Quantitative annotations**.  *Abundance versus anchor: (AVA)* For all prevalent metabolic features, mean abundance in each phenotype is calculated. Then, in each module, the metabolic feature (or standard in an annotated-module) with the maximum mean abundance in any phenotype is chosen as the "anchor". The anchor of an unannotated module is the metabolic feature with the highest mean abundance in any phenotype. In an annotated-module with only one standard, the standard is the anchor. If an annotated-module contains multiple standards, the anchor metabolite for that module is the standard with the highest mean abundance in any phenotype. The AVA of each feature $f$ in a module $m$ is then calculated as:

$$AVA_f = \frac{\max(u_1, u_2, .. u_n)}{\bar{A}}$$

where $n$ is the number of phenotypes, $u$ is the mean abundance in a phenotype and $\bar{A}$ is the maximum mean abundance of the co-clustered anchor.

*Association with phenotype: q value* MACARRoN integrates: MaAsLin 2 (Mallick et al, 2021) to efficiently capture multivariable associations between prevalent metabolic features and sample phenotypes (clinical or environmental metadata). Each metabolite feature is tested independently using a linear regression model:

$$feature \sim X_1 + ... + X_i + (1|R_1) + ... + (1|R_j)$$

where $X$ and $R$ are fixed and random effects in the metadata, respectively. $X_1$ represents the categorical metadata variable containing case and control phenotypes of interest that are used for prioritization. Other effects are regressed-out. Upon specification of the "control" (reference) phenotype within $X_1$, FDR-adjusted $P$ values ($q$ values) are determined for each feature for all pairs of case-control comparisons. Benjamini–Hochberg correction is applied to nominal $P$ values to control the FDR for independent tests.

*Association with phenotype: effect size* For each feature $f$, for each "case" phenotype, the effect size (ES) is calculated as:

$$ES_f = u_{case} - u_{control}$$

where $u$ is the mean of the $\log_2$-transformed abundances in a phenotype.

**Integration of ranks and prioritization**.  All prevalent metabolic features are ranked according to their AVA, effect size, and $q$ value yielding three ranks. Then, the priority score for each feature is calculated as the harmonic mean of the percentiles of the three ranks:

$$S(f) = \frac{n}{\sum_{i=1}^{n} \frac{1}{P(R)}}$$

where, $S(f)$ is the priority score of a feature f, $n$ is 3, and $P(R)$ is the percentile value of the rank ($R$) of each property including AVA, effect size, and $q$ value. The MACARRoN package includes a utility —"showBest" to further refine prioritization results to show the top "$n$" features that are prioritized in each annotated-module. It also provides the user the $\Delta m/z$s of these features with respect to the anchor in that module, should they wish to filter for a biochemical transformation or mass-difference of interest.

### Bioactivity prioritization in inflammatory bowel disease

**HMP2 stool metabolomics**.  In all, 546 stool metabolomes were used with MACARRoN, drawn from the HMP2 cohort publicly available at the Inflammatory Bowel Disease Multi'omics Database and the Human Microbiome Bioactives Resource Portal (https://portal.microbiome-bioactives.org/) (Lloyd-Price et al, 2019). These included 265 Crohn's disease (CD), 146 ulcerative colitis (UC), and 135 non-IBD metabolomes collected from 102 participants followed for up to a year each. For each sample, relevant clinical information such as age, gender, disease severity (dysbiosis score), and antibiotic use were available. Each metabolome was analyzed by four LC–MS methods; (1) C18-neg: for targeting metabolites of intermediate polarity such as free fatty acids and bile acids, (2) C8-pos: for polar and nonpolar lipids, and (3) HILIC-neg, and (4) HILIC-pos: for polar metabolites (Lloyd-Price et al, 2019). In order to standardize within and across batches, nearest-neighbor scaling was done using the flanking pooled stool samples. Next, median normalization was performed in a per-metabolite, per-method manner using the total signal median for each sample in each method to account for water content and heterogeneity across stool samples.

**Classification of features**.  In total, 81,867 total ion features were initially available from the HMP2 metabolomes. For initial annotation of confidently identified compounds, we (i) matched recorded RTs and $m/z$s to mixtures of reference metabolites analyzed in each batch and (ii) matched to an internal database of >600 compounds characterized using Broad Institute protocols (Lloyd-Price et al, 2019). To assign mass-matches to remaining features, we (iii) matched $m/z$s to monoisotopic molecular weights of entries in HMBD 5.0 (Wishart et al, 2022). For (i) and (ii), confident annotation of 596 features was possible with these strategies and we refer to these as standards.

For (iii), mass-matching with HMDB entries, the following ionizations were considered for features detected using C8-pos and HILIC-pos: [M]+ (+0), [M+H]+ (+1.007825), [M+Na]+ (+22.98977), [M+NH4]+ (+18.03437), [M-H2O+H]+ (-17.00274). For features detected using C18-neg and HILIC-neg, they were: [M]-, [M-H]- (-1.007825), [M+FormicAcid-H]- (+44.9982026), [M+AceticAcid-H]- (+59.0127559). Feature $m/z$s were compared with an allowed deviation of 10 ppm. Note that we did not use mass-matching against HMDB results for assigning identities to any features or for biological interpretation. Such information was drawn only from standards, as compared with

possibly known (putative mass-match with entry in the HMDB) and completely novel (unknown) compounds.

**Consolidation of redundant ion features**. Features likely to represent exactly the same ion were consolidated within each LC–MS method, based on an initial very high correlation in abundances across all samples. Spearman rank correlation threshold of 0.85 was used for C8-pos and C18-neg and 0.8 for the HILIC methods. When more than one feature clustered above these thresholds, only the feature with the highest mean abundance was labeled as the primary feature. Features that co-eluted (ΔRT of 0.015 min for C8-pos and C18-neg and of 0.025 min for HILIC) with the primary feature were considered redundant. 14,634 (17.9%) were identified as redundant and represented adducts, multimers, or in-source ion fragments. These were omitted from subsequent analyses.

**Detection of modules**. Of the 67,233 primary ion features detected in the metabolomes, those present in at least 70% of the metabolomes of at least one phenotype (CD; UC; non-IBD) were selected for further analysis, yielding 37,201 features. Correlations between pairs of these features were calculated using $\log_2$-transformed abundances in each phenotype, and the highest of the three bicors was used for tree construction. Modules in the resulting dendrogram were identified using the default MACARRoN module detection parameters and minClusterSize of 33 ($\sqrt[3]{37{,}201}$).

**Quantitative annotations**. *AVA*: In each module, the metabolic feature with the highest mean abundance in any phenotype (CD; UC; non-IBD) was chosen as the anchor. For annotated-modules (modules with at least one standard), only the standards were candidates for anchors. AVA for a metabolic feature was the ratio of its highest mean abundance in any phenotype and that of the anchor, as described earlier.

*Effect size and* q *value*: Mixed-effect linear regression models were implemented using MaAsLin 2 v0.99.12 (Mallick et al, 2021) to identify metabolic features that were differentially abundant between non-IBD and IBD (CD and UC) as well as between dysbiosis states within each phenotype. First, zero values were additively smoothed by 1 on a per-feature basis (smallest non-zero value was 5). Then, the abundances of the 37,201 metabolic features were $\log_2$-transformed and fitted to a similar per-feature linear model as in the previous HMP2 analysis (Lloyd-Price et al, 2019).

$$\begin{aligned} feature \sim{}& diagnosis + diagnosis : dysbiosis \\ & + antibiotics + age + (1|subject) \end{aligned}$$

In other words, transformed abundances were modeled as a function of diagnosis (a categorical variable with non-IBD as the reference group) and dysbiosis states as a binary nested variable within each diagnosis category (phenotype) with nondysbiosis as the reference variable. Effects contributed by antibiotic usage and age of the subject were regressed-out. Since each subject contributed multiple samples, subject IDs were used as a random effect. Nominal *P* values were FDR-adjusted on a per-metadata, per-category basis using Benjamini–Hochberg to obtain *q* values. Highest number of significantly differentially abundant features were observed for contrasts (1) CD-dysbiosis and CD-nondysbiosis and (2) UC and non-IBD. Effect sizes for each feature were calculated as the

difference in means of $\log_2$-transformed abundances for these contrasts.

**Prioritization of IBD-associated bioactives**. Prioritization of potential bioactives was performed using CD-dysbiosis and UC as target disease arms. The 37,201 primary features were considered as candidates for prioritization. To prioritize potential bioactives associated with UC, all features were meta-ranked using ranks from AVAs, *q* values and effect sizes in UC (versus non-IBD). The same process was repeated for CD-dysbiosis. The highly prioritized in each target disease subtype were considered as potentially bioactive. Of these, we specifically focused on features co-clustered with a standard (characterizable features) as candidates for further computational and/or experimental analyses.

*Evaluation of modules*

**Estimation of homogeneity**. Chemical taxonomy (class and subclass) information for standard metabolites was downloaded from the HMDB using their HMDB accession IDs. For modules containing ≥2 standards, MACARRoN counted each unique class contained in that module and percent homogeneity (H) was calculated as:

$$H = \frac{\max(c_1, c, ..., c_n)}{S} \times 100$$

where *c* is the count of a class, *n* is the number of unique classes and S is the total number of standards.

**Comparison with GNPS clusters**. *Cystic fibrosis sputum metabolomics*: MS1 abundances for 9280 metabolic features in 101 sputum metabolomes included in a cross-sectional study of cystic fibrosis severity (Quinn et al, 2019) were kindly provided by Dr. Robert A. Quinn. Results of the GNPS molecular networks analysis for the same study were downloaded from gnps.ucsd.edu (MassIVE ID: MSV000080655). GNPS uses the cosine score, a measure of similarity between MS2 fragmentation spectra, to associate compounds in molecular networks. In other words, compounds in a GNPS molecular network are connected if the alignment of their fragmentation spectra yields a cosine score ≥0.7 (default). The GNPS molecular networks of the sputum metabolomes included 1938 metabolites; cosine scores and GNPS molecular network IDs were available for 2356 pairs of metabolites. All cosine scores were ≥0.7. Of the 1938 metabolites, MS1 abundances were available for 1311 (9280 MS1 ∩ 1938 MS2).

*Comparison between bicors and cosines* Sample metadata included 4 distinct "disease states" of the sputum donor: A (clinical decision to treat with intravenous antibiotics; *n* = 15), B (on treatment; *n* = 2), C (post treatment *n* = 11), and D (no treatment or clinical decision to treat; *n* = 65) (Quinn et al, 2019). Metabolomes of disease state B and without disease state information were removed from analysis. As per MACARRoN methodology, for all pairs of metabolites, bicors in each of the disease states were determined and the best was considered. Both bicor and cosine values were available for 1030 pairs of metabolites.

*ARI between GNPS clusters and MACARRoN modules* Best-observed bicor was considered for dendrogram construction. Modules were detected using default MACARRoN arguments and

minClusterSize of 21 ($\sqrt[3]{9280}$). The 9280 features were distributed among 210 modules and 132 singletons. For the 1311 metabolites common in both MS1 and MS2 datasets, similarity between the GNPS and MACARRoN cluster assignments (instances whether a pair of features had the same GNPS cluster ID and MACARRoN module ID) was calculated using adjusted Rand Index (ARI) with the mclust (Scrucca et al, 2016) R package. To test whether the overlap between GNPS and MACARRoN was non-random, MACARRoN module assignments were shuffled (×1000) and compared to GNPS (and vice versa ×1000) and ARI was determined for each comparison. Actual ARI and ARI from randomized-assignments were compared using a one-sample $t$ test.

*Analyzing mass-differences in annotated-modules* The $m/z$ values of all features were rounded off to the first decimal. The mass-differences ($\Delta m/z$s) between all pairs of unannotated metabolic features and co-clustered standards in each annotated module (M) were calculated as:

$$\Delta m/z = m/z^M_{unannotated} - m/z^M_{standard}$$

Distribution of $\Delta m/z$s in real versus random annotated-modules: The module labels were shuffled, and $\Delta m/z$s were calculated for each new module (M') that contained a random set of standards and unannotated metabolic features.

$$\Delta m/z' = m/z^{M'}_{unannotated} - m/z^{M'}_{standard}$$

This was repeated 1000 times, and then each of the 1000 $\Delta m/z$ distributions was compared to the distribution obtained from actual modules using the Kolmogorov–Smirnov test.

Enrichment of $\Delta m/z$s in real versus random modules: Another evaluation criterion we applied to the resulting modules was the distribution of $\Delta m/z$s, which we would expect to be enriched for small positive (and occasionally negative) deltas when they represent the addition (or removal) of functional groups from a parent backbone. In order to inspect these distributions, we grouped $\Delta m/z$s by one-unit bins for ease of analysis. Each bin's frequency was calculated, and all $\Delta m/z$ bins observed more than once were considered. In total, this yielded 694 positive and 499 negative $\Delta m/z$ bins. As described before, random modules were generated by shuffling labels, and the frequency of each of the 1193 $\Delta m/z$ bins was noted. This was repeated 10,000 times. An empirical $P$ value of enrichment for each $\Delta m/z$ bin was calculated as (North et al, 2002):

$$p = \frac{r+1}{n+1}$$

where $n$ is the total number of iterations i.e., 10,000, and $r$ is the number of iterations in which the frequency of the $\Delta m/z$ bin was greater than or equal to that calculated from the real modules.

### Confirmation of prioritized bilirubin/stercobilin pathway members from modules 10 and 85 in MassIVE molecular networks

**Classical molecular networking workflow**. In support of MACARRoN's predicted bilirubin pathway derivatives, we identified 11 MassIVE datasets corresponding to MS2 analysis of human stool

metabolomes (Dataset EV9). For each dataset, a molecular network was created using the default online workflow (https://ccms-ucsd.github.io/GNPSDocumentation/) on the GNPS website (http://gnps.ucsd.edu). The data was filtered by removing all MS2 fragment ions within +/− 17 Da of the precursor $m/z$. MS2 spectra were window-filtered by choosing only the top six fragment ions in the +/− 50 Da window throughout the spectrum. The precursor ion mass tolerance was set to 2.0 Da and a MS/MS fragment ion tolerance of 0.5 Da. A network was then created where edges were filtered to have a cosine score above 0.7 and more than 6 matched peaks. Further, edges between two nodes were kept in the network if and only if each of the nodes appeared in each other's respective top 10 most similar nodes. Finally, the maximum size of a molecular family was set to 100, and the lowest-scoring edges were removed from molecular families until the molecular family size was below this threshold. The spectra in the network were then searched against GNPS' spectral libraries. The library spectra were filtered in the same manner as the input data. All matches kept between network spectra and library spectra were required to have a cosine score above 0.7 and at least 6 matched peaks.

**Identifying matches between features in modules 10 and 85 and stercobilin molecular networks**. Of the molecular networks identified for each of the 11 MassIVE datasets, those containing features annotated as stercobilin were selected. Then, features that were directly connected to stercobilin and within $\Delta m/z$ of 100 w.r.t to it were considered. If the stercobilin molecular networks contained a feature(s) annotated as urobilin, its neighbors satisfying the aforementioned criteria were also considered. The $m/z$s of these features were then compared to the $m/z$s of highly prioritized features in modules 10 and 85 that were within $\Delta m/z$ of 100 of stercobilin.

### Host vs. microbial NAD enzyme activity in NR metabolism

**Differential abundance of microbial NAD biosynthesis enzymes in IBD**. Perturbation statuses of *nadA* and *nadB* (*de novo*) and *nadR*, *pncA*, and *pncB* (salvage) protein families (UniRef90) were obtained from a previous study (Zhang et al, 2022). Briefly, abundances of each protein family were estimated from 1595 metagenomes from 130 participants in HMP2 (CD, $n = 65$; UC, $n = 38$; non-IBD, $n = 27$) using MetaWIBELE v0.3.8. The abundance values were normalized to copies per million (CPM) units and log-transformed. Zero values were additively smoothed by half of the smallest non-zero measurement on a per-feature basis. The same random effects model formulation applied to HMP2 metabolomics was applied here within MaAsLin 2 (Mallick et al, 2021), which estimated the differential abundance of each protein family in dysbiotic (active) versus non-dysbiotic CD and UC.

**Transcription of host NAD biosynthesis enzymes**. The host transcriptomes (HTX) from the HMP2 dataset were downloaded from http://ibdmdb.org in July 2020. To match HTX samples with MBX samples, we considered the first pair of MBX:HTX samples from each subject that were separated by no more than 2 weeks (yielding 71 HTX samples from 71 individuals: 33 with CD, 21 with UC, and 17 non-IBD controls). We performed normalization on raw sample-by-gene count data of the selected HTX samples using the voom method implemented in R's limma package (Law et al, 2014;

Ritchie et al, 2015). The normalized counts were used as a measure of gene expression.

### Testing NR's effects on colonic injury and inflammation in mouse models using DSS

**Mice.** C57BL/6J (B6) mice were purchased from Jackson Laboratory and were housed in the Harvard T.H. Chan School of Public Health. All mice were housed in microisolator cages in the barrier facility with a constant 12-h light cycle. Female mice were used at 6–8 weeks of age and fed standard chow (PicoLab Mouse Diet 5058). Mice were randomized to experimental groups one week prior to the start of an experimental intervention to minimize cage-based or housing bias. All animal studies and experiments described in this manuscript were approved and carried out in accordance with the Harvard Medical School's Standing Committee on Animals and the National Institutes of Health guidelines for animal use and care.

**Acute DSS-induced colonic injury and inflammation model.** C57BL/6J WT were treated with 3% (w/v) DSS ad libitum in the drinking water for 5 days and followed by regular drinking water for 2 days. Body weight was measured every day and mice were sacrificed on day 7. On sacrifice, colon length was measured and then fixed with 4% paraformaldehyde for histology. NR (1000 mg/kg) or PBS intraperitoneal injections started 2 days before DSS was added to the drinking water and continued until the day of sacrifice at the same time daily.

**Chronic DSS-induced colonic injury and inflammation model.** C57BL/6J WT were treated with three cycles of DSS as follows: cycle 1 (3% (w/v) DSS ad libitum in the drinking water for 5 days and followed by regular drinking water for 9 days), cycle 2 (2% (w/v) DSS ad libitum in drinking water for 5 days and followed by regular drinking water for 9 days), and cycle 3 (1% (w/v) DSS ad libitum in drinking water for 5 days and followed by regular drinking water for 2 days). On sacrifice, colon length was measured, and colons were opened longitudinally prior to fixation with 4% paraformaldehyde (PFA). NR (1000 mg/kg) (Brown et al, 2014; Igarashi et al, 2019) or PBS intraperitoneal injections started 2 days before initiation of the first cycle of DSS and continued until the day of sacrifice at the same time daily. Body weight was measured every day until the mice were sacrificed.

**Histology.** Colons were cleaned with PBS prior to fixation in 4% PFA and then processed by routine paraffin embedding, sectioning, and hematoxylin and eosin (H&E) staining for the acute colitis model or both H&E and Masson's trichrome staining for the chronic DSS model. Colitis scores were determined by a pathologist (JNG), who was blinded to the experimental parameters. Each of the four histologic parameters was scored as absent (0), mild (1), moderate (2), or severe (3): mononuclear cell infiltration, polymorphonuclear cell infiltration, epithelial hyperplasia, and epithelial injury. The scores for the parameters were summed to generate the histologic colitis score and were further quantified to include the percentage involvement by the disease process: (1) <10%; (2) 10–25%; (3) 30–50%; (4) >50% and presented as histologic colitis scores as follows: cumulative score * % involvement (Chun et al, 2019).

### Quantifying highly prioritized stercobilin-linked compounds in mouse metabolomes

**Mice.** Germ-free (GF) WT C57BL/6J mice were bred and maintained in isocages under a strict 12-h light cycle in the Harvard T. H. Chan Gnotobiotic Center for Mechanistic Microbiome studies. WT C57BL/6J (SPF) mice were purchased from Jackson Laboratory (Bar Harbor, Maine) and were maintained under the same conditions as GF mice for 3 weeks. Mice used for the experiment were between 8 and 9 weeks of age and fed sterilizable soy protein-free extruded rodent diet (Envigo CAT #: 2020SX).

**Metabolomics.** Fecal contents (30–100 mg) were collected from both GF and SPF mice and snap-frozen. Stool samples were processed and untargeted metabolomics data were collected using the HILIC-positive LC–MS method as previously described (Lloyd-Price et al, 2019).

## Analyzed data

The HMP2 stool metabolomics and human transcriptomics datasets analyzed in this manuscript have been retrieved from https://portal.microbiome-bioactives.org/ and http://ibdmdb.org, respectively, and are referenced in "Reagents and Tools table and Methods and Protocols".

## Data availability

The MACARRoN tool developed in this work is available as a Bioconductor package (https://bioconductor.org/packages/Macarron). Metabolic features prioritized as potentially bioactive in CD-dysbiosis, and UC are provided in Dataset EV7. Code for application of MACARRoN on the HMP2 metabolomics dataset, downstream analyses, and figures in the manuscript is available at https://github.com/biobakery/macarron_manuscript.git.

## Peer review information

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

## Acknowledgements

This work was partially supported by NIH NIDDK grants R24DK110499 (CH, WSG, and RJX). We especially appreciate the participants in the HMP2 cohort who made this study possible. The computations in this paper were run in part on the FASRC Cannon cluster supported by the FAS Division of Science Research Computing Group at Harvard University. We thank Dana-Farber/Harvard Cancer Center for the use of the Rodent Histopathology Core which processed intestinal tissue for the assessment of histopathology.

## Author contributions

**Amrisha Bhosle**: Conceptualization; Data curation; Software; Formal analysis; Validation; Visualization; Methodology; Writing—original draft; Writing—review and editing. **Sena Bae**: Resources; Formal analysis; Validation; Investigation; Visualization; Writing—original draft; Writing—review and editing. **Yancong Zhang**: Software; Formal analysis; Writing—review and editing. **Eunyoung Chun**: Investigation. **Julian Avila-Pacheco**: Resources; Data curation; Writing—review and editing. **Ludwig Geistlinger**: Software. **Gleb Pishchany**: Investigation. **Jonathan N Glickman**: Investigation. **Monia Michaud**: Investigation. **Levi Waldron**: Supervision. **Clary B Clish**: Resources; Supervision. **Ramnik J Xavier**: Supervision; Funding acquisition. **Hera Vlamakis**: Data curation. **Eric A Franzosa**: Conceptualization; Supervision; Methodology; Writing—review and editing. **Wendy S Garrett**: Resources; Supervision; Funding acquisition; Project administration; Writing—review and editing. **Curtis Huttenhower**: Conceptualization; Supervision; Funding acquisition; Methodology; Project administration; Writing—review and editing.

## Disclosure and competing interests statement

CH is on the scientific advisory board of ZOE, Seres Therapeutics, and Empress Therapeutics. WSG is on the scientific advisory board of Freya Biosciences, Sail Biomedicines, Scipher Medicine, and Empress Therapeutics. The laboratory of WSG receives funding from Merck and Astellas, unrelated to current work. RJX is a member of the scientific advisory board of Nestle and Senda Biosciences.

# Expanded View Figures

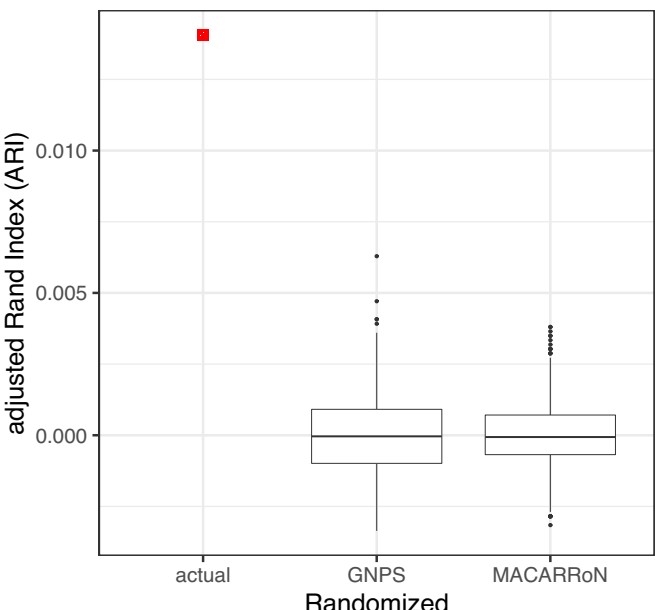

**Figure EV1.   Adjusted Rand Index (ARI) values for the overlap between actual and randomized MACARRoN modules and GNPS clusters (molecular networks).**

Overlap between GNPS cluster and MACARRoN module assignments of 1311 features in the sputum metabolomes was determined using ARI. GNPS cluster and MACARRoN module assignments were then shuffled and overlap with actual MACARRoN modules and GNPS clusters respectively was calculated. This was performed 1000 times each for both GNPS and MACARRoN. The ARI obtained by comparing actual assignments was significantly higher than ARIs where one assignment was randomized. Bounds of boxplots show 1$^{st}$ quartile and 3$^{rd}$ quartile and the line inside the box is the median.

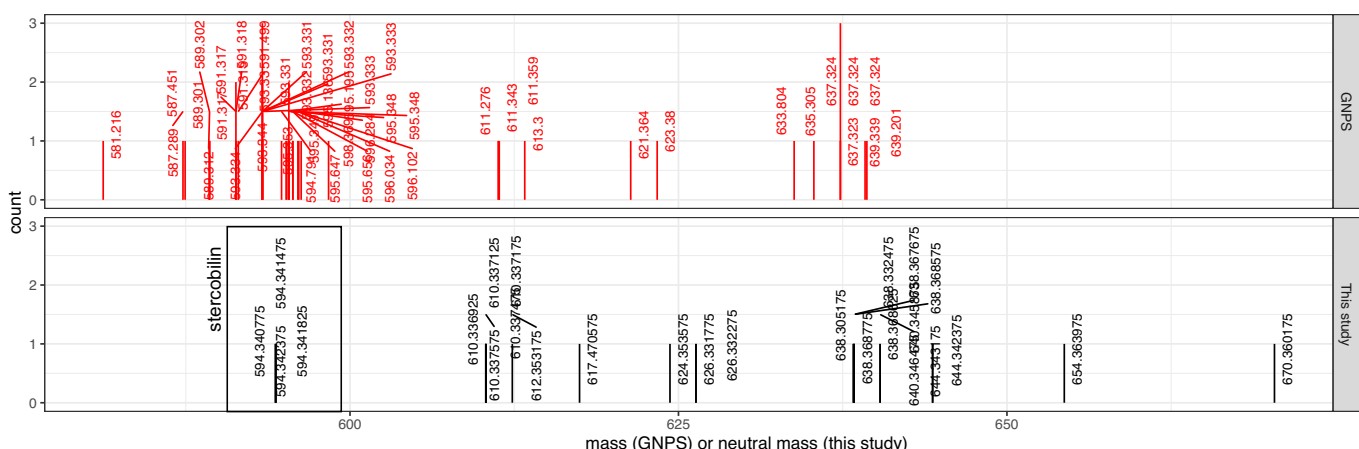

**Figure EV2. Masses of highly prioritized features in modules 10 and 85, and compounds in stercobilin molecular networks across 11 MassIVE datasets.**

(bottom) Shown are the highly prioritized features in modules 10 and 85 with |Δ*m/z*| <100 w.r.t stercobilin. (top) Compounds that were directly connected to stercobilin or urobilin in molecular networks within the aforementioned mass-difference shown. The tight cluster of masses represents masses associated with stercobilin in these molecular networks. Count represents the number of times a particular mass was observed.

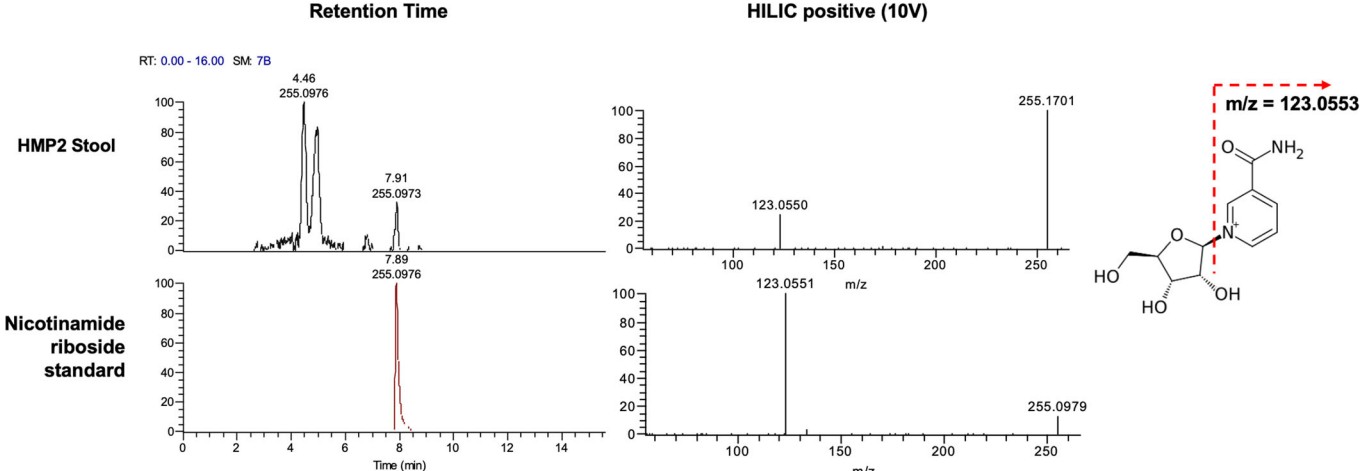

**Figure EV3.  Identification of feature representing nicotinamide riboside in the HMP2.**

Retention time and fragmentation pattern on HILIC-positive are shown for HMP2 feature and nicotinamide riboside standard.

