## [Peer Review File · Molecular Systems Biology]

Integrated annotation prioritizes metabolites with bioactivity in inflammatory bowel disease

Amrisha Bhosle, Sena Bae, Yancong Zhang, Eunyoung Chun, Julian Avila-Pacheco, Ludwig Geistlinger, Gleb Pishchany, Jonathan Glickman, Monia Michaud, Levi Waldron, Clary Clish, Ramnik Xavier, Hera Vlamakis, Eric Franzosa, Wendy Garrett, and Curtis Huttenhower

Corresponding author(s): Curtis Huttenhower (chuttenh@hsph.harvard.edu) , Wendy Garrett (wgarrett@hsph.harvard.edu), Eric Franzosa (franzosa@hsph.harvard.edu)

Review Timeline:

Submission Date:	20th Sep 23
Editorial Decision:	16th Nov 23
Revision Received:	31st Jan 24
Editorial Decision:	6th Feb 24
Revision Received:	13th Feb 24
Accepted:	15th Feb 24

Editor: Maria Polychronidou

Transaction Report:

16th Nov 2023

Manuscript Number: MSB-2023-12024

Title: Integrated annotation prioritizes metabolites with bioactivity in inflammatory bowel disease

Dear Curtis,

Thank you again for submitting your work to Molecular Systems Biology. My apologies once again for the delay in sending you our editorial decision. As I already mentioned, we unfortunately have not managed to obtain a report from reviewer #1, despite numerous reminders. We have now heard back from reviewers #2 and #3. As you will see below, both reviewers find the study interesting and relevant for the microbiome community. They do however raise a series of concerns, which we would ask you to address in a revision.

The comments of the referees are rather clear and seem straightforward to address so I think there is no need to repeat any of them here. All issues raised by the referees would need to be satisfactorily addressed. Please let me know in case you would like to discuss in further detail any of the issues raised, I would be happy to schedule a call.

On a more editorial level, we would ask you to address the following points:

- Please provide a .doc version of the manuscript text (including legends for the main figures) and individual production quality figure files for the main Figures (one file per figure).
- Please include 5 keywords.
- We have replaced Supplementary Information by the Expanded View (EV format). In this case, all additional figures can be provided as EV Figures. Please provide one file per EV Figure. Their legends should be included in the manuscript text. For detailed instructions regarding expanded view please refer to our Author Guidelines: .
- Tables S1-S12 should be provided and called out in the text as Datasets EV1-EV12. Please provide one file per EV Dataset. Please include the description of each EV Dataset in the dataset file itself, i.e. in a separate tab for .xls files or as a README.txt file in .zip folders for .csv files.
- Please provide a "standfirst text" summarizing the study in one or two sentences (approximately 250 characters), three to four "bullet points" highlighting the main findings and a "synopsis image" (550px width and max 400px height, jpeg format) to highlight the paper on our homepage.
- Please include a "Disclosure and Competing Interests statement" in the main text.
- All Materials and Methods need to be described in the main text. We would encourage you to use 'Structured Methods', our new Materials and Methods format. According to this format, the Material and Methods section should include a Reagents and Tools Table (listing key reagents, experimental models, software and relevant equipment and including their sources and relevant identifiers) followed by a Methods and Protocols section in which we encourage the authors to describe their methods using a step-by-step protocol format with bullet points, to facilitate the adoption of the methodologies across labs. More information on how to adhere to this format as well as downloadable templates (.doc or .xls) for the Reagents and Tools Table can be found in our author guidelines: . An example of a Method paper with Structured Methods can be found here:
- Please include a Data availability section describing how the data, code etc. have been made available. This section needs to be formatted according to the example below:
The datasets and computer code produced in this study are available in the following databases:
 - Chip-Seq data: Gene Expression Omnibus GSE46748 (<https://www.ncbi.nlm.nih.gov/geo/query/acc.cgi?acc=GSE46748>)
 - Modeling computer scripts: GitHub (<https://github.com/SysBioChalmers/GECKO/releases/tag/v1.0>)
 - [data type]: [full name of the resource] [accession number/identifier] ([doi or URL or identifiers.org/DATABASE:ACCESSION])
- For data quantification: please specify the name of the statistical test used to generate error bars and P values, the number (n) of independent experiments (specify technical or biological replicates) underlying each data point and the test used to calculate p-values in each figure legend. The figure legends should contain a basic description of n, P and the test applied. Graphs must include a description of the bars and the error bars (s.d., s.e.m.).
- The References should be formatted according to the Molecular Systems Biology reference style (i.e., ordered alphabetically and listing the first 10 authors followed by et al).

- When you resubmit your manuscript, please download our CHECKLIST (<https://bit.ly/EMBOPressAuthorChecklist>) and include the completed form in your submission.

Please note that the Author Checklist will be published alongside the paper as part of the transparent process (<https://www.embopress.org/page/journal/17444292/authorguide#transparentprocess>).

If you feel you can satisfactorily deal with these points and those listed by the referees, you may wish to submit a revised version of your manuscript. Please attach a covering letter giving details of the way in which you have handled each of the points raised by the referees. A revised manuscript will be once again subject to review and you probably understand that we can give you no guarantee at this stage that the eventual outcome will be favorable.

Kind regards,

Maria

Maria Polychronidou, PhD
Senior Editor
Molecular Systems Biology

We realize that it is difficult to revise to a specific deadline. In the interest of protecting the conceptual advance provided by the work, we recommend a revision within 3 months (14th Feb 2024). Please discuss the revision progress ahead of this time with the editor if you require more time to complete the revisions. Use the link below to submit your revision:

IMPORTANT: When you send your revision, we will require the following items:

1. the manuscript text in LaTeX, RTF or MS Word format
2. a letter with a detailed description of the changes made in response to the referees. Please specify clearly the exact places in the text (pages and paragraphs) where each change has been made in response to each specific comment given
3. three to four 'bullet points' highlighting the main findings of your study
4. a short 'blurb' text summarizing in two sentences the study (max. 250 characters)
5. a 'thumbnail image' (550px width and max 400px height, Illustrator, PowerPoint or jpeg format), which can be used as 'visual title' for the synopsis section of your paper.
6. Please include an author contributions statement after the Acknowledgements section (see <https://www.embopress.org/page/journal/17444292/authorguide>)
7. Please complete the CHECKLIST available at (<https://bit.ly/EMBOPressAuthorChecklist>).
Please note that the Author Checklist will be published alongside the paper as part of the transparent process (<https://www.embopress.org/page/journal/17444292/authorguide#transparentprocess>).

See also figure legend guidelines: <https://www.embopress.org/page/journal/17444292/authorguide#figureformat>

9. Please note that corresponding authors are required to supply an ORCID ID for their name upon submission of a revised manuscript (EMBO Press signed a joint statement to encourage ORCID adoption). (<https://www.embopress.org/page/journal/17444292/authorguide#editorialprocess>)

Currently, our records indicate that there is no ORCID associated with your account.

Please click the link below to provide an ORCID:

Link Not Available

*** PLEASE NOTE *** As part of the EMBO Press transparent editorial process initiative (see our Editorial at <https://dx.doi.org/10.1038/msb.2010.72>), Molecular Systems Biology publishes online a Review Process File with each accepted manuscripts. This file will be published in conjunction with your paper and will include the anonymous referee reports, your point-by-point response and all pertinent correspondence relating to the manuscript. If you do NOT want this File to be

published, please inform the editorial office at msb@embo.org within 14 days upon receipt of the present letter.

Reviewer #2:

Bhosle et al. introduce the package MACARRoN which seeks to illuminate the chemical origins of the tens of thousands of unannotated hits from untargeted fecal metabolomics datasets by grouping them into modules that are likely to be chemically related. It does so by examining covariance and size similarity patterns in relation to known standards. Using this package, they compare healthy controls to inflammatory bowel disease subjects and make observations consistent with literature and also find novel associations involving vitamin metabolites. The authors find that one such metabolite, nicotinamide riboside, is depleted in IBD and is protective in animal models of acute and chronic colitis. In summary, the authors present a unique and highly useful tool for the microbiome community and demonstrates its value in the use-case of inflammatory bowel disease.

Comments:

The manuscript states that the samples queried came from "gut metabolomes from the 114 Integrative Human Microbiome Project (iHMP or HMP2) spanning 411 IBD patients (265 Crohn's 115 disease, 146 ulcerative colitis) and 135 non-IBD control individuals". The HMP2 included many fewer individuals but these individuals were followed longitudinally. Do the authors mean to state those numbers as the number of samples (likely some from the same patients), instead of patients? Lines 134-137 suggest this is the case.

The comparison in lines 138-140 is a Welch's test, but to my knowledge this test cannot account for multiple samples from the same individual. Could the authors perform statistical tests that allow one to control for intra-individual covariance such as linear mixed effects as in later comparisons?

Lines 174-176 state "Taken together, the prevalence, abundance, and phenotype-associated perturbation of the 12,020 unannotated features pointed towards the richness of *microbe*-associated potential bioactives in the IBD gut metabolome." As the authors allude to in lines 58-61, the presence of metabolites in feces alone does not unambiguously indicate they are microbe-derived compounds, as host metabolites from serum access this space. The comparison by the authors of germfree to SPF mice is quite valuable and perhaps could be used to illuminate in a comprehensive fashion the compound modules that are modified by the microbiota, akin to the comparison in Figure 4F. Mouse and human hosts produce different compounds (e.g. primary bile acids), but it may still be informative to see for the metabolomics hits that are found in both mice and humans, which individual compounds or what proportion of each module are dependent on the presence of gut microbes (i.e. are differentially abundant in germfree compared to SPF mice). This could be an interesting lens from which to examine the compounds that associate with IBD. Providing this information as a column in a Supplemental Table that comprises all the detected metabolomic features could also be quite useful to the community that will use Macarron.

Additionally, might the authors be able to utilize public metabolomics datasets from broad-spectrum antibiotic-treated humans to identify which metabolites in their dataset are linked to the presence of microbes?

Reviewer #3:

The paper by Bhosle et al. describes a new tool, MACARRoN, that prioritizes metabolites from complex microbiome samples that are potentially associated with a phenotype of interest, and applies this to inflammatory bowel disease. The paper is well-written, and the method is thoroughly put together and looks solid. What is particularly nice about the method is that it combines several different types of approaches, such as statistical association analysis and analysis of mass-spectral data.

While I am generally supportive of the paper, I do have a number of points of feedback that I hope will be useful to the authors to consider:

- I find the use of the term 'bioactive' in the paper confusing. In lines 69-70 it is defined as 'either causally or responsively involved in a [...] phenotype', but in general I believe the field would interpret it the term as being linked to causation ('active', instead of passive, denotes this too). Given that many (the large majority of?) prioritized molecules are likely not actively/causally involved in mediating the phenotype but rather are passively covarying with the phenotype, I believe that the use of this term is confusing and undesired. This is all the more the case given the compositionality of the data and the large reduction of microbiome complexity in IBD samples versus healthy ones. I would therefore recommend going for a more neutral term such as 'associated with the phenotype'.

- MACARRoN identifies whether metabolites are similar to covarying known metabolites. This is shown to enrich for cases where the mass difference amounts to a chemically defined distance, such as a methyl group or an oxidation. I wonder whether it would make sense to (at least allow for) specifically filter(ing) the similar-metabolite connections to only those with chemically defined mass differences, potentially corroborated by molecular formula predictions.

- The integration with spectral networking is nice. Could spectral similarity (above a certain threshold) be included in the

network/module visualizations, using a distinct edge type? In this way one could both see the statistical associations and the putative structural associations.

- Line 66: besides BioTransformer, tools like MetWork/CANPA could also be mentioned here.

Response to reviewers

for

Integrated annotation prioritizes metabolites with bioactivity in inflammatory bowel disease

Amrisha Bhosle^{1,2,3}, Sena Bae⁴, Yancong Zhang^{1,2,3}, Eunyoung Chun⁴, Julian Avila-Pacheco⁵,
Ludwig Geistlinger^{6,7}, Gleb Pishchany¹, Jonathan N. Glickman^{8,9}, Monia Michaud⁴, Levi
Waldron⁶, Clary Clish⁵, Ramnik J. Xavier^{1,10,11}, Hera Vlamakis¹, Eric A. Franzosa^{1,2,3*}, Wendy S.
Garrett^{1,3,4,12*}, Curtis Huttenhower^{1,2,3,4*}

¹Infectious Disease and Microbiome Program, Broad Institute of MIT and Harvard, Cambridge, MA, USA

²Department of Biostatistics, Harvard T. H. Chan School of Public Health, Boston, MA, USA

³Harvard Chan Microbiome in Public Health Center, Harvard T. H. Chan School of Public Health, Boston, MA, USA

⁴Department of Immunology and Infectious Diseases, Harvard T. H. Chan School of Public Health, Boston, MA, USA

⁵Metabolomics Platform, Broad Institute of MIT and Harvard, Cambridge, MA, USA

⁶Department of Epidemiology and Biostatistics, Graduate School of Public Health and Health Policy, City University of New York, New York, NY, USA

⁷Center for Computational Biomedicine, Harvard Medical School, Boston, MA, USA

⁸Beth Israel Deaconess Medical Center, Boston, MA, USA

⁹Department of Pathology, Harvard Medical School, Boston, MA, USA

¹⁰Gastrointestinal Unit and Center for the Study of Inflammatory Bowel Disease, Massachusetts General Hospital and Harvard Medical School, Boston, MA, USA

¹¹Center for Microbiome Informatics and Therapeutics, Massachusetts Institute of Technology, Cambridge, MA, USA

¹²Department of Medical Oncology, Dana-Farber Cancer Institute, Boston, MA, USA

* These authors jointly supervised this work.

Reviewer 2:

Bhosle et al. introduce the package MACARRoN which seeks to illuminate the chemical origins of the tens of thousands of unannotated hits from untargeted fecal metabolomics datasets by grouping them into modules that are likely to be chemically related. It does so by examining covariance and size similarity patterns in relation to known standards. Using this package, they compare healthy controls to inflammatory bowel disease subjects and make observations consistent with literature and also find novel associations involving vitamin metabolites. The authors find that one such metabolite, nicotinamide riboside, is depleted in IBD and is protective in animal models of acute and chronic colitis. In summary, the authors present a unique and highly useful tool for the microbiome community and demonstrates its value in the use-case of inflammatory bowel disease.

We thank the reviewer for their time and encouraging comments on the study.

Comments:

2.1. The manuscript states that the samples queried came from "gut metabolomes from the 114 Integrative Human Microbiome Project (iHMP or HMP2) spanning 411 IBD patients (265 Crohn's 115 disease, 146 ulcerative colitis) and 135 non-IBD control individuals". The HMP2 included many fewer individuals but these individuals were followed longitudinally. Do the authors mean to state those numbers as the number of samples (likely some from the same patients), instead of patients? Lines 134-137 suggest this is the case.

The reviewer is correct that those are sample and not participant numbers, and we apologize for the confusion caused by the typographical error. The information in lines 134-137 (pre-revision), i.e. "These metabolomes were profiled from 265 Crohn's disease (CD), 146 ulcerative colitis (UC) and 135 non-IBD stool samples from 106 participants, followed longitudinally for up to one year each," was correct. We have made the correction to (lines 126-129):

"We thus developed a method to prioritize potentially bioactive metabolomic features from microbial communities, using it to explore 546 single-MS (MS1) gut metabolomes from the Integrative Human Microbiome Project (iHMP or HMP2) spanning 80 IBD patients and 26 non-IBD (control) individuals that were longitudinally profiled."

2.2 The comparison in lines 138-140 is a Welch's test, but to my knowledge this test cannot account for multiple samples from the same individual. Could the authors perform statistical tests that allow one to control for intra-individual covariance such as linear mixed effects as in later comparisons?

We thank the reviewer for pointing this out. We have revised this calculation to account for repeated measurements i.e. multiple metabolomes from a participant and now compare the number of features per participant (instead of per metabolome) across diagnoses. The number of features associated with a participant is calculated as the average number of features detected in their metabolomes. Our conclusions remain unchanged. We have revised the text in the manuscript accordingly to (lines 153-156; please note that we have also rounded values in this subsection to their nearest integers for improved readability):

"Participants with IBD (n = 80) contributed fewer features than non-IBD controls (n = 26) on average (CD (n = 50): 49957, UC (n = 30): 49511, and non-IBD: 50,910, Welch's two-sample t-test, CD vs. non-IBD p -value = 0.037 and UC vs. non-IBD p -value = 0.016), consistent with reduced metabolite (and microbiome) diversity in IBD".

We have also confirmed this finding using a linear mixed effects model: (feature count ~ diagnosis, random = ~1|participant), which revealed significant differences between the number of features in CD vs. non-IBD (p -value = 0.044) and

UC vs. non-IBD (p -value = 0.0154). In the interest of simplicity, we have opted to use the corrected Welch's t-test in the revised manuscript to support our finding.

*2.3. Lines 174-176 state "Taken together, the prevalence, abundance, and phenotype-associated perturbation of the 12,020 unannotated features pointed towards the richness of *microbe*-associated potential bioactives in the IBD gut metabolome." As the authors allude to in lines 58-61, the presence of metabolites in feces alone does not unambiguously indicate they are microbe-derived compounds, as host metabolites from serum access this space. The comparison by the authors of germfree to SPF mice is quite valuable and perhaps could be used to illuminate in a comprehensive fashion the compound modules that are modified by the microbiota, akin to the comparison in Figure 4F. Mouse and human hosts produce different compounds (e.g. primary bile acids), but it may still be informative to see for the metabolomics hits that are found in both mice and humans, which individual compounds or what proportion of each module are dependent on the presence of gut microbes (i.e. are differentially abundant in germfree compared to SPF mice). This could be an interesting lens from which to examine the compounds that associate with IBD. Providing this information as a column in a Supplemental Table that comprises all the detected metabolomic features could also be quite useful to the community that will use Macarron.*

Reviewer 2 is correct that the presence of metabolites in stool alone does not unambiguously indicate that they are microbe-derived, and comparison of IBD-associated metabolites to those enriched in SPF vs. germ-free mice may shed light on their potential microbial origin. (Note: In this regard, we would also like to mention that MACARRoN's implementation and prioritization is agnostic to the source or origin of a metabolite – we have now made this clearer in the Discussion (lines 583-587). The method's application to the HMP2 dataset is a special case where prioritized features are expected to include microbially-derived metabolites since disrupted microbial activity is central to IBD.)

The metabolomics data that allowed us to find the overlap between the prioritized stercobilin-associated compounds and compounds that are enriched in SPF mice (Fig. 4F) was graciously provided by our collaborator for the sole purpose of confirming the microbial origins of the stercobilin-associated metabolites as a part of this study. The broader comparison of other human/IBD-associated and SPF metabolites will be published along with our collaborator's study in the near future.

In the meantime, several previous studies (Han *et al.*, 2021; Marcobal *et al.*, 2013; Meier *et al.*, 2023) on SPF vs germ-free stool metabolomes, and databases such as Exposome Explorer (http://exposome-explorer.iarc.fr/microbial_metabolites) (Neveu *et al.*, 2023) can also be leveraged to infer metabolites that interact with the microbiome, and we have listed them in the Discussion in lines 542-549:

“Despite the small overlap between the number of prioritized and validated bilirubin-associated compounds (previous studies (Han *et al.*, 2021; Marcobal *et al.*, 2013; Meier *et al.*, 2023; Neveu *et al.*, 2023) may be used to infer the broader overlap between (IBD-associated) human and SPF-specific metabolites belonging to other chemical classes), the findings are still notable given many factors that differentiate this system from human IBD: lower gut microbial diversity in SPF mice, markedly different diets as compared to humans, absence of inflammation as in dysbiotic human subjects, and thus, potential absence of host-derived, inflammation-linked precursor substrates.”

2.4 Additionally, might the authors be able to utilize public metabolomics datasets from broad-spectrum antibiotic-treated humans to identify which metabolites in their dataset are linked to the presence of microbes?

We thank the reviewer for this suggestion. Changes in the stool metabolome following antibiotic exposure have indeed been previously observed in mice (Antunes *et al*, 2011; Choo *et al*, 2017; Vrbanac *et al*, 2020; Zhang *et al*, 2022a) and human infants (Patton *et al*, 2020; Russell *et al*, 2021). Unfortunately, we did not find publicly available adult human stool metabolomics datasets to perform this analysis with MACARRoN. A recent study (Folz *et al*, 2023) that profiles the intestinal and stool metabolomes of 15 healthy human adults includes only two participants with antibiotic exposure, which is insufficient for the correlation and effect size calculations in MACARRoN. The aforementioned database, Exposome Explorer (Neveu *et al.*, 2023) includes 457 microbial metabolites with evidence of perturbation from antibiotic exposure or germ-free vs SPF metabolomics studies. In any case, MACARRoN prioritizes all metabolites that are linked with a phenotype of interest such as antibiotic exposure. The prioritized metabolites could either be microbe-derived or be associated with other microbiome-independent physiological processes during antibiotic exposure; it would require additional biochemical assays (just like the SPF vs. GF measurements that we refer to for the prioritized stercobilin-associated compounds) to confirm their microbial origin.

Reviewer 3:

The paper by Bhosle et al. describes a new tool, MACARRoN, that prioritizes metabolites from complex microbiome samples that are potentially associated with a phenotype of interest, and applies this to inflammatory bowel disease. The paper is well-written, and the method is thoroughly put together and looks solid. What is particularly nice about the method is that it combines several different types of approaches, such as statistical association analysis and analysis of mass-spectral data.

While I am generally supportive of the paper, I do have a number of points of feedback that I hope will be useful to the authors to consider:

We are grateful to the reviewer for their positive feedback on our method and manuscript.

3.1 I find the use of the term 'bioactive' in the paper confusing. In lines 69-70 it is defined as 'either causally or responsively involved in a [...] phenotype', but in general I believe the field would interpret the term as being linked to causation ('active', instead of passive, denotes this too). Given that many (the large majority of?) prioritized molecules are likely not actively/causally involved in mediating the phenotype but rather are passively covarying with the phenotype, I believe that the use of this term is confusing and undesired. This is all the more the case given the compositionality of the data and the large reduction of microbiome complexity in IBD samples versus healthy ones. I would therefore recommend going for a more neutral term such as 'associated with the phenotype'.

We recognize that the term “bioactive” may provoke an assumption of causality for some readers. In this work, we use the term in an inclusive manner to indicate prioritized features that are enriched for metabolites that are causal, responsive, or incidentally associated (i.e. ‘passively covarying’) with the phenotype. Each of these subtypes of association have potential diagnostic, therapeutic, or microbiome functional characterization applications. In addition, our prediction of metabolite bioactivity includes but is not limited to ‘association with the phenotype’, as prevalence, covariance, and abundance with respect to characterized metabolites are also important criteria. We have now made this clearer in the revised manuscript by defining the term “bioactive” in the lines 71-74 in Introduction and lines 560-563 in Discussion:

Introduction: “We define metabolite bioactivity as its causal, responsive, or incidental association with a health, disease, or exposure response phenotype - each of these associations have potential diagnostic, metabolic, or (microbiome) functional characterization applications.”

Discussion: “It (MACARRoN) also combines these and other more basic properties such as prevalence and abundance with respect to a known metabolite with phenotypic “importance” in prioritization of potential bioactivity, i.e. the likelihood of its causal or consequential involvement in a phenotype or an environment of interest.”

Additionally, our use of the term “bioactive” in this work is geared towards maintaining consistency with published works from our group. More specifically, MACARRoN was developed under the umbrella of the Human Microbiome Bioactives Resource (HMBR) (<https://www.microbiome-bioactives.org/>), which lists species, strains, genes, and metabolites under “potentially bioactive elements of the gut microbiome.” In addition, MACARRoN’s “sister” method in the HMBR, MetaWIBELE, which is aimed at prioritizing microbial gene families rather than metabolites, employed a similarly broad definition of “potentially bioactive” in its recent publication (Zhang *et al*, 2022b) to encompass various mechanisms of feature-phenotype association and ecological importance. That being said, we are willing to make the change to a more neutral term if the reviewer continues to feel strongly about the use of “bioactive” in this work.

3.2 MACCARRoN identifies whether metabolites are similar to covarying known metabolites. This is shown to enrich for cases where the mass difference amounts to a chemically defined distance, such as a methyl group or an oxidation. I wonder whether it would make sense to (at least allow for) specifically filter(ing) the similar-metabolite connections to only those with chemically defined mass differences, potentially corroborated by molecular formula predictions.

We have included a utility (`showBest`) in the MACARRoN package (<https://bioconductor.org/packages/Macarron>) that provides users $\Delta m/z$ s between unannotated prioritized metabolic features and covarying annotated metabolites (i.e. anchors in the same module) in the column titled “m.z_vs_Anchor,” should they wish to filter for a transformation of interest (now included in the Methods; lines 700-704). A snippet of the file generated with this utility is shown below:

Compound	HMDB.ID	Metabolite	m.z	m.z_vs_Anchor	Priority_score	Status	Module	Anchor	AVA	q-value	effect_size
C8p_QI45	HMDB08925*	C34:0 PE	720.5558	0	0.4097	depleted in CD.dys	22	C34:0 PE	1	0.75574355	0.8215
C8p_QI22527			698.5717	-21.984	0.9877	depleted in CD.dys	22	C34:0 PE	4.1889	0.00012303	5.4792
C8p_QI21222			670.5404	-50.015	0.9846	depleted in CD.dys	22	C34:0 PE	8.4869	0.00014221	4.5599
C8p_QI4923			359.2944	-361.261	0.9822	depleted in CD.dys	22	C34:0 PE	1.4376	0.00055518	8.1474
C8p_QI21680			680.5611	-39.995	0.9812	depleted in CD.dys	22	C34:0 PE	1.7045	0.00093068	6.8259
C8p_QI20622			657.5451	-63.011	0.9803	depleted in CD.dys	22	C34:0 PE	1.5834	0.00049335	5.746
C8p_QI21305			672.556	-48	0.9782	depleted in CD.dys	22	C34:0 PE	12.3151	0.00019922	3.9213

While mass differences that correspond to well-defined biochemical transformations or chemical groups are surely more informative, we do not pre-filter results based on these to avoid excluding (a) associations between compounds that are a result of two or more step biochemical transformations (note that this can be any combination of well-defined transformations) or (b) associations that indicate completely novel one-step transformations. Another reason to not pre-filter based on well-defined mass differences is that the observed $\Delta m/z$ s are affected by accuracy of the instrument, as well as presence of adducts and fragments.

3.3 The integration with spectral networking is nice. Could spectral similarity (above a certain threshold) be included in the network/module visualizations, using a distinct edge type? In this way one could both see the statistical associations and the putative structural associations.

We thank the reviewer for their positive feedback on the integration of spectral networking. This is an excellent suggestion and we have revised Fig. 4D to include edges representing structural similarity. Briefly, in the GNPS molecular networks of 4 MassIVE datasets, stercobilin was seen to be spectrally similar (i.e. cosine score > 0.7) to 4 compounds, the m/z s of which were similar to those of unannotated prioritized compounds in our dataset (please see below a snippet of Table EV9 with the relevant information). We have now added edges between stercobilin and these 3 very-highly prioritized (priority score > 0.9) metabolic features that are mass-adjacent (or “overlapping” with) to compounds in the GNPS molecular networks. Please note that given the lack of fragmentation data in our study, it is impossible to derive 1:1 correspondence between the unannotated compounds in our study and those in the GNPS molecular network.

MassIVE dataset	Stercobilin Neighbor ID in MassIVE	Neighbor m/z	Anchor m/z	Edge Cosine	Edge $\Delta m/z$ (w.r.t. anchor)	m/z of overlapping very-highly prioritized (PS > 0.9) feature in this study	Feature ID of overlapping very-highly prioritized (PS > 0.9) feature in this study
MSV000082262	29061	639.201	594.761	0.9	44.44	639.3403 (85); 639.3766 (10)	HILp_QI22686 (85); HILp_QI22689 (10)
MSV000079651	2204	639.339	595.349	0.8	43.99	639.3403 (85); 639.3766 (10)	HILp_QI22686 (85); HILp_QI22689 (10)
MSV000079777	348286	611.276	593.765	0.87	17.511	611.3453 (10)	HILp_QI21471 (10)
MSV000082629	66436	611.343	595.597	0.82	15.746	611.3453 (10)	HILp_QI21471 (10)

Revised Fig. 4D: Pink dashed edges represent structural similarity between stercobilin and features in the GNPS molecular networks that are mass-adjacent to (i.e. overlapping with) the top prioritized compounds in modules 10 and 85. For example, the edge between HILp_QI22686 and stercobilin represents the structural similarity between stercobilin and features 29061 (MSV000082262) and 2204 (MSV000079651) that are overlapping with HILp_QI22686. Labels represent cosine scores provided in Table EV9.

3.4 Line 66: besides BioTransformer, tools like MetWork/CANPA could also be mentioned here.

We thank the reviewer for the suggestion and have included this reference in this sentence (line 603).

References

- Antunes LC, Han J, Ferreira RB, Lolic P, Borchers CH, Finlay BB (2011) Effect of antibiotic treatment on the intestinal metabolome. *Antimicrob Agents Chemother* 55: 1494-1503
- Choo JM, Kanno T, Zain NM, Leong LE, Abell GC, Keeble JE, Bruce KD, Mason AJ, Rogers GB (2017) Divergent Relationships between Fecal Microbiota and Metabolome following Distinct Antibiotic-Induced Disruptions. *mSphere* 2
- Folz J, Culver RN, Morales JM, Grembi J, Triadafilopoulos G, Relman DA, Huang KC, Shalon D, Fiehn O (2023) Human metabolome variation along the upper intestinal tract. *Nat Metab* 5: 777-788
- Han S, Van Treuren W, Fischer CR, Merrill BD, DeFelice BC, Sanchez JM, Higginbottom SK, Guthrie L, Fall LA, Dodd D *et al* (2021) A metabolomics pipeline for the mechanistic interrogation of the gut microbiome. *Nature* 595: 415-420
- Marcobal A, Kashyap PC, Nelson TA, Aronov PA, Donia MS, Spormann A, Fischbach MA, Sonnenburg JL (2013) A metabolomic view of how the human gut microbiota impacts the host metabolome using humanized and gnotobiotic mice. *ISME J* 7: 1933-1943
- Meier KHU, Trouillon J, Li H, Lang M, Fuhrer T, Zamboni N, Sunagawa S, Macpherson AJ, Sauer U (2023) Metabolic landscape of the male mouse gut identifies different niches determined by microbial activities. *Nat Metab* 5: 968-980
- Neveu V, Nicolas G, Amara A, Salek RM, Scalbert A (2023) The human microbial exposome: expanding the Exposome-Explorer database with gut microbial metabolites. *Sci Rep* 13: 1946
- Patton L, Li N, Garrett TJ, Ruoss JL, Russell JT, de la Cruz D, Bazacliu C, Polin RA, Triplett EW, Neu J (2020) Antibiotics Effects on the Fecal Metabolome in Preterm Infants. *Metabolites* 10
- Russell JT, Lauren Ruoss J, de la Cruz D, Li N, Bazacliu C, Patton L, McKinley KL, Garrett TJ, Polin RA, Triplett EW *et al* (2021) Antibiotics and the developing intestinal microbiome, metabolome and inflammatory environment in a randomized trial of preterm infants. *Sci Rep* 11: 1943
- Vrbanac A, Patras KA, Jarmusch AK, Mills RH, Shing SR, Quinn RA, Vargas F, Gonzalez DJ, Dorrestein PC, Knight R *et al* (2020) Evaluating Organism-Wide Changes in the Metabolome and Microbiome following a Single Dose of Antibiotic. *mSystems* 5
- Zhang N, Liu J, Chen Z, Chen N, Gu F, He Q (2022a) Integrated Analysis of the Alterations in Gut Microbiota and Metabolites of Mice Induced After Long-Term Intervention With Different Antibiotics. *Front Microbiol* 13: 832915
- Zhang Y, Bhosle A, Bae S, Mclver LJ, Pishchany G, Accorsi EK, Thompson KN, Arze C, Wang Y, Subramanian A *et al* (2022b) Discovery of bioactive microbial gene products in inflammatory bowel disease. *Nature* 606: 754-760

6th Feb 2024

Manuscript Number: MSB-2023-12024R

Title: Integrated annotation prioritizes metabolites with bioactivity in inflammatory bowel disease

Dear Curtis,

Thank you for sending us your revised manuscript. We have now heard back from reviewer #2 who was asked to evaluate your revised study. As you will see below, the reviewer is satisfied with the performed revisions and supports publication. As such, I am glad to inform you that your study can soon be accepted for publication, pending some editorial issues listed below, which we would ask you to address in a minor revision.

- Our Data Editors noted that the following needs to be corrected/added in the Figure Legends:

-- Please indicate the statistical test used for data analysis in the legend of figure 2e.

-- In figures 4e-f; there is a mismatch between the annotated p values in the figure legend and the annotated p values in the figure file that should be corrected."

-- The box plots need to be defined in terms of minima, maxima, centre, bounds of box and whiskers, and percentile in the legends of figures 5c-d.

-- The box plots need to be defined in terms of minima, maxima, and whiskers in the legends of figures 4e-f; EV 1.

-- Information related to n is missing in the legends of figures 5c-d; EV 1.

-- Although 'n' is provided, please describe the nature of entity for 'n' (biological? technical?) in the legend of figure 4f."

- The funding information provided in the manuscript text (Acknowledgements) should match the information entered in the online submission system. Currently "FAS Division of Science Research Computing Group at Harvard University" is missing from the submission system.

- There is a callout for Fig. 2H in the text, but the panel is missing.

- Tables EV1-EV10 should be uploaded individually (one file per EV Dataset). They need to be renamed to Dataset EV1-EV10 and their callouts in the text should be updated.

- The Appendix file should be removed and the "Note" should be incorporated in the text (e.g. in the Discussion where it is referenced).

Please resubmit your revised manuscript ****within one month**** and ideally as soon as possible. If we do not receive the revised manuscript within this time period, the file might be closed and any subsequent resubmission would be treated as a new manuscript. Please use the Manuscript Number (above) in all correspondence.

Click on the link below to submit your revised paper.

Kind regards,

Maria

Maria Polychronidou, PhD
Senior Editor
Molecular Systems Biology

If you do choose to resubmit, please click on the link below to submit the revision online before 7th Mar 2024.

IMPORTANT:

Please note that corresponding authors are required to supply an ORCID ID for their name upon submission of a revised manuscript (EMBO Press signed a joint statement to encourage ORCID adoption).

(<https://www.embopress.org/page/journal/17444292/authorguide#editorialprocess>)

Currently, our records indicate that the ORCID for your account is 0000-0002-1110-0096.

Link Not Available

*** PLEASE NOTE *** As part of the EMBO Press transparent editorial process initiative (see our Editorial at <https://dx.doi.org/10.1038/msb.2010.72> , Molecular Systems Biology will publish online a Review Process File to accompany accepted manuscripts. When preparing your letter of response, please be aware that in the event of acceptance, your cover letter/point-by-point document will be included as part of this File, which will be available to the scientific community. More information about this initiative is available in our Instructions to Authors. If you have any questions about this initiative, please contact the editorial office (msb@embo.org).

Reviewer #2:

I appreciate the authors' responses to comments and believe that this manuscript constitutes a significant advance in the field.

All editorial and formatting issues were resolved by the authors.

15th Feb 2024

Manuscript number: MSB-2023-12024RR

Title: Integrated annotation prioritizes metabolites with bioactivity in inflammatory bowel disease

Dear Curtis,

Thank you again for sending us your revised manuscript. We are now satisfied with the modifications made and I am pleased to inform you that your paper has been accepted for publication.

Kind regards,

Maria

Maria Polychronidou, PhD
Senior Editor
Molecular Systems Biology
